# Crossing chasms: A PLS approach to EU public sector performance assessment

Adriana AnaMaria Davidescu[1,2☯], Oana-Ramona Lobonț[3☯], Eduard Mihai Manta[4☯], Lavinia Daniela Mihiț[5☯], Alexandra-Mădălina Țăran[3☯] *

1 Department of Statistics and Econometrics, Bucharest University of Economic Studies, Bucharest, Romania, 2 National Scientific Research Institute for Labour and Social Protection, Bucharest, Romania, 3 Finance, Business Information Systems and Modelling Department, Faculty of Economics and Business Administration, West University of Timisoara, Timisoara, Romania, 4 Doctoral School of Cybernetics and Statistics, Bucharest University of Economic Studies, Bucharest, Romania, 5 Doctoral School of Economics and Business Administration, West University of Timisoara, Timisoara, Romania

☯ These authors contributed equally to this work.
* alexandra.taran@e-uvt.ro

## Abstract

This paper examines the dynamics of public sector performance across European Union (EU) countries through a comprehensive methodological framework. This study introduces the European Public Sector Performance Index, a novel approach that employs Partial Least Squares (PLS) econometric modelling and cluster analysis to evaluate public sector performance from 2007 to 2021. By assessing performance across governance, social, and economic dimensions, the research captures the multifaceted nature of public sector efficiency in the EU. Our investigation reveals significant determinants of performance, including governance factors like Control of Corruption, Rule of Law, and Government Effectiveness, as well as economic indicators such as Inflation and social factors like Equity of access to healthcare services and Education Spending. These findings underscore the critical role of transparent governance, economic stability, and equitable social policies in enhancing public sector efficiency. Despite its reliance on secondary data and the PLS method, the study provides new methodological insights and empirical evidence on public sector performance, contributing to the literature with a holistic analysis that integrates digitalisation and well-being. This study's holistic approach offers actionable insights for policymakers and stakeholders, emphasising the need for robust governance and equitable policies to improve public sector performance across the EU. The omission of certain societal components—such as economic conditions, demographic changes, or cultural factors—may result in a skewed representation of how digital transformation and governance interact. These external factors can significantly influence the effectiveness of digital initiatives and the overall performance of public institutions.

**Data availability statement:** All relevant data are within the manuscript and its Supporting Information files.

**Funding:** The research study has been elaborated within the Data Science Research Lab for Business and Economics of the Bucharest University of Economic Studies, within the project funded by the European Union's NextGenerationEU instrument through Romania's National Recovery and Resilience Plan - Pillar III-C9-I8, managed by the Ministry of Research, Innovation, and Digitalization, as part of the project titled "CauseFinder: Causality in the Era of Big Data and AI and its Applications in Innovation Management," contract no. 760049/23.05.2023, code CF 268/29.11.2023. The funders had no role in study design, data collection and analysis, decision to publish, or preparation of the manuscript.

**Competing interests:** The authors have declared that no competing interests exist.

## Introduction

The main purpose of this paper is to investigate the public sector performance across the European Union countries, incorporating various dimensions of governance, service delivery, and socioeconomic and digital transformation within a complex methodological framework.

The European Union member states have intensified efforts to enhance public sector performance, triggering a series of reforms and strategic initiatives aimed at administrative efficiency and innovation. Amidst societal and digital transformations, the public sector faces pressures to adapt and innovate, addressing challenges such as efficiency, digital disruption, well-being, and health. This evolving landscape underscores the importance of developing precise performance measurement indicators and robust empirical evidence to refine public sector policies and strategies [1–7]. The complexity of these challenges, amplified by the global COVID-19 pandemic and economic uncertainties, necessitates advanced methodological approaches to capture the multifaceted nature of public sector performance. Research highlights the potential of comprehensive evaluations and the application of innovative econometric models, such as Partial Least Squares (PLS), to assess public sector efficiency [8–10].

However, gaps remain in integrating new dimensions—digitalisation and well-being—into performance evaluations and applying advanced analytical techniques to provide actionable insights [2,11].

Our study aims to bridge these gaps by employing a holistic approach to evaluate public sector performance within European Union countries. We introduce the European Public Sector Performance Index, leveraging bibliometric analysis, content analysis, and empirical research. Specifically, we utilise the Partial Least Squares (PLS) method and cluster analysis to assess performance from 2007 to 2021 across governance, social, and economic dimensions, using data from 27 EU countries. This approach allows us to systematically explore and identify performance trends, contributing to the literature with new methodological insights and empirical evidence on public sector performance.

Central to our investigation is the overarching research question: "How can the performance of the public sector in European Union countries be accurately measured and analysed across diverse dimensions, including governance, social, and economic aspects, approaching a novel methodological framework?" This inquiry aims to dissect the multifaceted nature of public sector performance by integrating traditional and emergent dimensions such as digitalisation and well-being. By employing an innovative methodological approach encompassing PLS econometric modelling and cluster analysis, our study seeks to develop a comprehensive understanding of the performance dynamics within EU countries. The research question guides our exploration towards identifying and applying a robust set of indicators and methods capable of capturing nuanced performance variations across the EU's diverse public sectors. Through this question, we aim to contribute to the extant literature by offering empirical insights and methodological advancements in evaluating public sector performance, specifically within the context of the European Union.

Our research examines the dynamics of public sector performance across European Union (EU) countries, introducing the European Public Sector Performance Index, a novel approach that employs Partial Least Squares (PLS) econometric modelling and cluster analysis. Overall, our results reveal the dynamics of public sector performance across European Union (EU) countries. The research results capture the multifaceted nature of public sector efficiency by assessing performance across governance, social, and economic dimensions. Thus, our main results underscore the importance of robust governance and equitable policies to enhance public sector performance across the EU, providing valuable insights for policymakers. The composite index constructed across the pillars demonstrated the considerable effects of governance factors, respectively the control of corruption, the protection and promotion of human rights, and the effectiveness of the governments, having considerable importance within the model and emphasising that the transparent and accountable governance structures can further sustain the increase of public sector's overall performance. Notably, the most significant inferences in fostering effective governance come from a significant control of corruption, while the boost of economic health, social welfare, and robust governance can be achieved through equal access to healthcare services, alongside an increase in the level of financing as regards the education sector. Additionally, our results highlighted that the digital economy acquires low importance in the governance pillar due to a lack of regard for the decision-makers in implementing efficient digital initiatives within public governance.

The research presented in this study offers several novel contributions to the existing specialised literature on public sector performance measurement. The uniqueness of our findings stems from both the diversity of indicators used and the innovative methodologies employed, which collectively enhance our understanding of public sector dynamics, particularly within the context of the EU-27 Member States. While existing literature often constructs public sector performance indices that focus on a limited set of relative pillars, our study fills a significant gap by demonstrating how multiple dimensions can be integrated into a cohesive measurement framework. This multidimensional approach allows for a more nuanced understanding of public sector performance, recognising that traditional indicators alone may not fully capture the complexities of contemporary governance challenges. Second, the application of complex empirical methods further distinguishes our research. By utilising composite indicators at the level of each pillar (governance, economic, and social), we are able to assess the cumulative effects of various indicators. This methodological innovation enables us to construct a more comprehensive overall index of public sector performance. Additionally, our use of cluster analysis facilitates the identification of specific groups of countries based on their performance levels, allowing us to pinpoint dimensions that could significantly enhance public sector outcomes. This approach highlights country-specific stances and regional differences within the EU, thus contributing to tailored policy recommendations for improving performance. Third, the application of complex empirical methods further distinguishes our research. By utilising composite indicators at the level of each pillar (governance, economic, and social), we are able to assess the cumulative effects of various indicators. This methodological innovation enables us to construct a more comprehensive overall index of public sector performance. Additionally, our use of cluster analysis facilitates the identification of specific groups of countries based on their performance levels, allowing us to pinpoint dimensions that could significantly enhance public sector outcomes. This approach highlights country-specific stances and regional differences within the EU, thus contributing to tailored policy recommendations for improving performance. The fourth significant novel element of our research is the proposal for a dashboard designed to monitor and continuously update public sector performance measurement. This initiative, referred to as the European Public Sector Performance Index, aims to function as an observatory within the EU-27 Member States. By providing real-time data and insights, the dashboard will facilitate ongoing evaluations of public sector performance, promoting transparency and accountability while enabling stakeholders to make informed decisions based on up-to-date information. Five, another crucial contribution of our study is the detailed examination of public sector performance levels across European countries. Our findings reveal significant national differences that must be addressed to enhance the overall public sector effectiveness. By highlighting these disparities, we underscore the importance of developing targeted strategies to improve performance in specific contexts, thus fostering greater equity in public service delivery across the EU. Despite the study's reliance on secondary data and

the Partial Least Squares (PLS) method, it provides valuable methodological insights and empirical evidence concerning public sector performance. By integrating digitalisation and well-being into the analysis, we contribute a holistic perspective that enriches the existing literature and encourages future research to adopt similar comprehensive approaches.

The paper is organised as follows: Section 2 elaborates on the significance of the study and theoretical frameworks, including neo-institutionalism and behavioural theory, which underpin our research approach. Section 3 details the research methodology, describing the data, variables, and analytical techniques, including the PLS econometric model and cluster analysis. Section 4 presents our findings, revealing performance trends and variations across EU member states. Finally, based on our comprehensive assessment, we conclude with policy recommendations and strategies for enhancing public sector performance. This structure ensures a logical flow from conceptual underpinnings to empirical analysis, culminating in actionable insights for policymakers and stakeholders. Fig 1 summarises the paper structure.

Improving efficiency, boosting economic growth, and stimulating social well-being have proven to be key determinant factors in the performance of public sectors exposed to the increasing pressures of providing quality services. Thus, the performance investigation offers the opportunity to identify some recommendations for improving the performance related to various areas of the public sector through the adoption and implementation of the most effective practices and strategies through which the countries of the European Union can face and respond to different socio-economic, political or environmental challenges.

Therefore, numerous studies have depicted the importance of public sector performance and its measurement in many European Union member states and not only focusing on the different applied methodologies to construct composite indicators by exploring, testing, and providing various proxies for the measurement of the public sector performance and efficiency, along with sheer implications and recommendations.

The state of knowledge was investigated through comprehensive and in-depth bibliometric analysis to reveal the most important, prolific, and productive authors and the most highly cited documents in the field of research. Moreover, we also include other essential articles obtained through a systematic review and a series of scientific papers from the established authors concerned with our research topic by identifying a path that led us to other relevant articles in the existing literature on public sector performance.

Based on the fact that the existing literature is ample and complex, we propose to identify the state of the knowledge in the research field and to offer a detailed analysis and complex forthcoming perspectives that could be considered as a starting point for future research in this field regarding public sector performance and efficiency in terms of indicators and measurement. Thus, in order to achieve our research objective and to both measure and review the existing literature, a specific approach was applied based on the advanced methods of reviewing the specialised literature, such as those applied by other authors [12–14], consisting in conducting a bibliometric analysis.

Moreover, a more objective assessment of relevant documents can be achieved through the quantitative orientation [15,16]. Thus, significant productivity is represented by a high number of publications, on the one hand, and a high impact means a high number of citations for these documents, on the other hand [17–19]. Besides, this path can identify the most prolific and productive authors and cited documents [20]. Besides, through this path, some authors evidenced that the most prolific and productive authors and cited documents can be identified [20].

Making an investigation on the approach and inclusion of the concept of public sector performance and efficiency, we applied a bibliometric analysis that consisted of 189 articles extracted from the Clarivate Analytics, Web of Science Core Collection database for the last five years (2018–2022), implying two types of features (units of analysis), namely citations, and authors, respectively one type of analyses, more specific co-citation to map the literature and appraise the most productive authors graphically and to assess the most relevant documents, and cited references.

The final sample of documents encloses only articles, with a specific focus on those written in English to maintain the predominant scientific language, on the one hand, and the included period, on the other hand, with the dataset extracted for the 2018–2022 lapse of time.

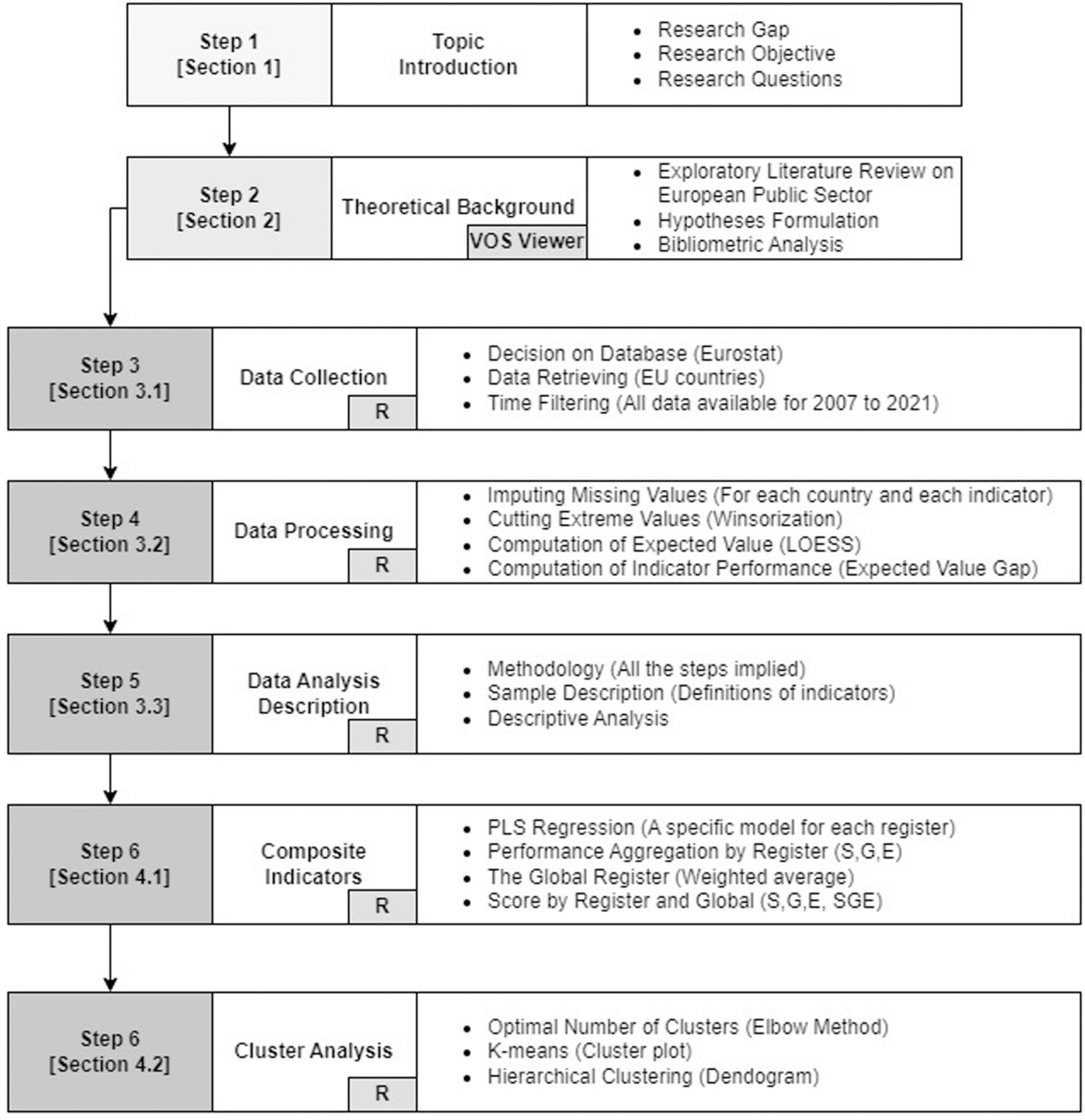

**Fig 1. Summary of paper structure.**

The design of the bibliometric analysis is detailed in Fig 2.

Co-citation represents a qualitative analysis, an initial approach that allows us to identify the most frequently referenced authors in the research field. Respectively, it refers to the fact that two authors are co-cited together by the same

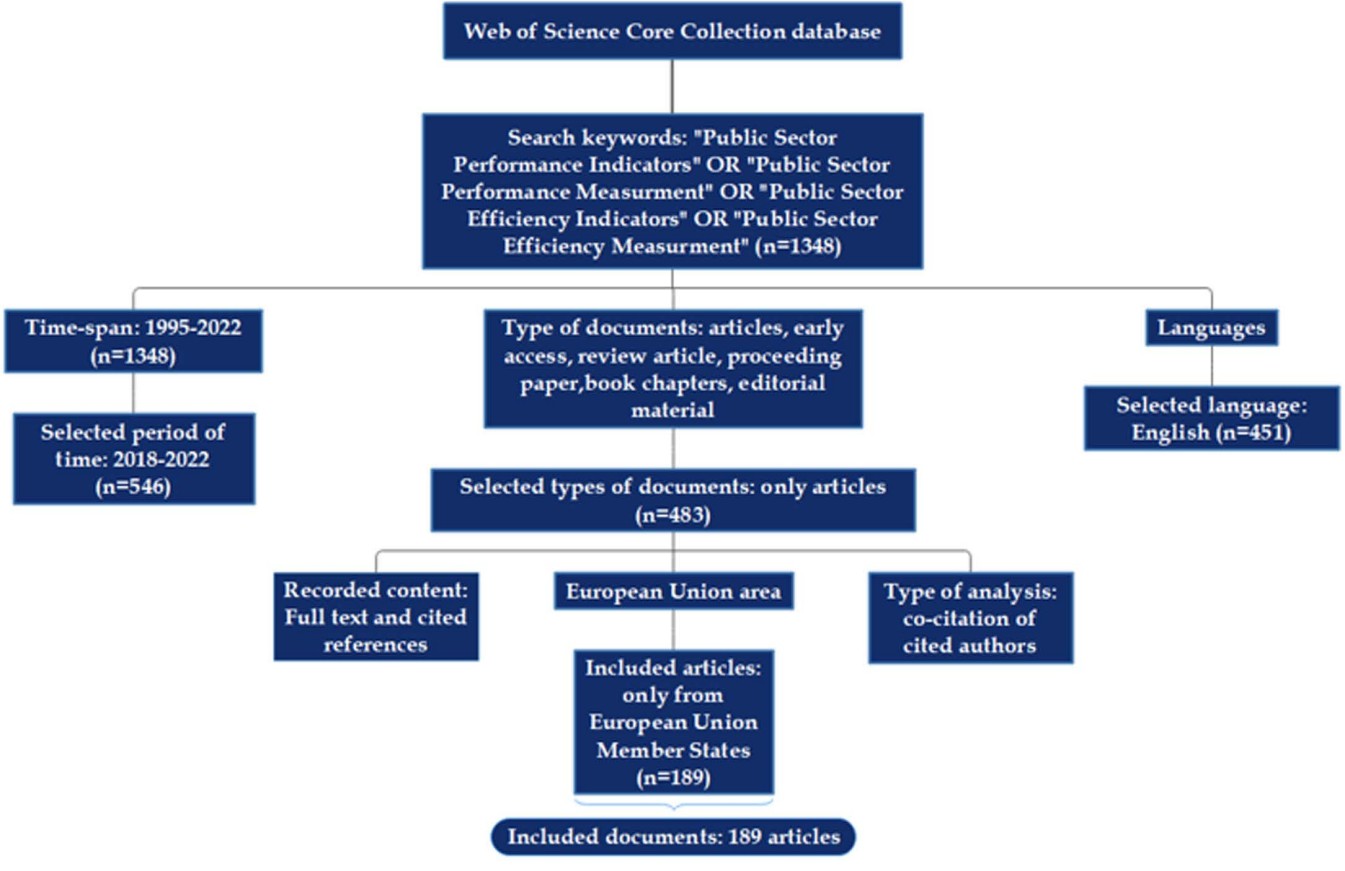

**Fig 2. The multi-phase process of determining the final sample of included articles.**

publication and are both in the reference list of the same article as the first authors [21]. The evolution of the top authors in a scientific field can be observed by co-citation analysis, referring to the frequency with which two articles are cited together in the third article [22]. The primary purpose is represented by extracting the most cited authors, identifying the main clusters, and graphically representing the results. Furthermore, Fig 3 maps the intellectual structure of the research field by visualising the co-citation network of cited authors, which helped us to identify the most cited authors and the most relevant articles in the research field to comprise them in our study.

The authors' co-citation network is graphically represented in Fig 3. Moreover, the analysis investigates and focuses on the research network by considering VOSviewer software, namely co-citation analysis in terms of cited authors, to stay connected to current developments and guide future research efforts. Likewise, the results of the scientific map have identified 8255 cited authors in the initial stage of the research. For a proper analysis, the threshold enclosed only the cited authors that have received, over the analysed period, a minimum number of 18 citations/author. Further, as the network threshold was configured and consisted of a specific number of citations, the final sample of authors captured only thirty cited authors that met the eligibility criteria of inclusion. The authors' co-citation analysis is designed from a dual presumption: the network of the most relevant and cited authors. The visualisation of the scientific research map highlights three distinct clusters: (i) cluster 1 (red) – which contains ten authors; (ii) cluster 2 (green) – which gathers ten authors; (iii) cluster 3 (blue) – with ten authors. Thus, it can be attested that each of the obtained clusters is formed robustly respectively, each item (author) belongs only to one cluster, the authors being equally distributed for each cluster based on the links

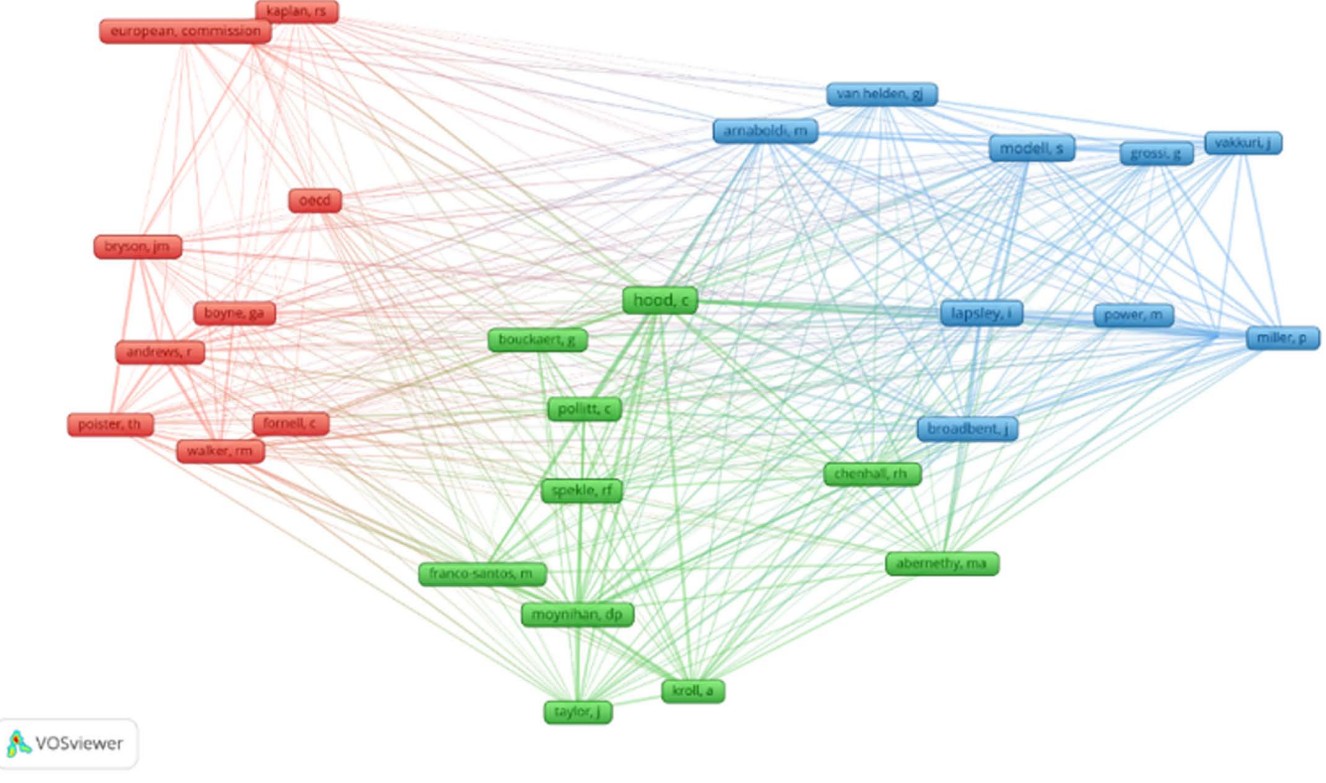

**Fig 3. Co-citation network of cited authors.**

(underline the intensity of the connection/collaboration of two authors) between them, along with the total link strengths (evidence the number of links of a cited author with other cited author, respectively the co-citation links of cited authors).

Within this frame, the obtained clusters with the most cited authors are pointed out in Table 1.

The results indicated that cluster 1 (red) is conducted by two key world institutions, OECD and the European Commission, with the authors with the highest number of citations, OECD with 30 citations, and the European Commission registered 27 citations. Furthermore, these international organisations are highly interested in the research field regarding public sector performance.

Withal, in the context of identifying a path to assess the performance of the public sector effectively, OECD [23] conducted a study to analyse the performance by focusing on four key characteristics as follows: (i) a proper manner to measure both effectiveness and efficiency of the public sector; (ii) the financial performance; (iii) saving resources; (iv) public sector quality. OECD [24] supports the development of a necessary comparison framework by dividing inputs, outputs, processes and outputs.

Moreover, we also notice the existence of a significant contribution to the field regarding the public sector performance by the European Commission. Thus, the European Commission has addressed, discussed, and proposed many research topics that cover distinct areas, including the public sector.

Cluster 2 (green) is led by the author Hood C, which has received 56 citations. Performance measurement is essential for the new public management initiatives, which have been developed to enhance the public sector's efficiency and effectiveness [25]. Moreover, international organisations such as the OECD and the World Bank consider performance measurement a well-discussed term after Hood's contribution [25]. Likewise, Cluster 3 (blue) comprises two influential authors

**Table 1. Clusters of the most cited authors.**

| No. | Cluster 1 (red) Author/ Citations | Cluster 2 (green) Author/ Citations | Cluster 3 (blue) Author/ Citations |
|---|---|---|---|
| 1 | OECD/30 | Hood, C/56 | Lapsley, I/43 |
| 2 | European Commission/27 | Pollitt, C/38 | Modell, S/42 |
| 3 | Andrews, R/22 | Moynihan, DP/33 | Arnaboldi, M/32 |
| 4 | Fornell, C/20 | Spekle, RF/31 | Broadbent, J/27 |
| 5 | Boyne, GA/20 | Bouckaert, G/23 | Miller, P/26 |
| 6 | Halplan, RS/20 | Chenhall, RH/21 | Grossi, G/22 |
| 7 | Walker, RM/20 | Kroll, A/20 | Vakkuri, J/21 |
| 8 | Bryson, JM/19 | Abernethy, MA/ 20 | Kurunmaki, L/19 |
| 9 | Nuti, S/18 | Franco-Santos, M/18 | Power, M/18 |
| 10 | Poister, TH/18 | Taylor, J/18 | Van Helden, GJ/19 |

with the highest citations (Lapsley – 43 citations and Modell – 42 citations). Modell discussed the new developments in institutional research on performance measurement in the public sector [26]. Along the same lines, associated challenges regarding the development and implementation of performance indicators are well established by many studies [27].

Since performance measurement is a complex process, it requires appropriate consideration of many related sub-domains when computing composite indicators. Nevertheless, we have revealed through the bibliometric analysis the most important international organisations and cited authors, such as the European Commission, OECD, Hood C, Pollit C, Lapsley I, and Modell S, that are considered the most cited authors and the generators of specific research frameworks, applying distinct methodologies and different indicators that can be included when we conduct a comparative international analysis between countries in terms of public sector performance.

Considering that the final sample of documents contains only articles, exclusively those indexed in the Web of Science, we thought relevant other international institutions that provide official statistics and that include other authors, a fact that led to the expansion of information and studies related to the performance of the public sector by identifying and considering other relevant documents specific to the existing framework of the literature.

As the groundwork for the credibility initiate process only to the Web of Science database and understanding that relevant studies on public sector performance could be omitted, we continue our approach through content analysis, gathering all the identified relevant scientific documents in Table 2.

The groundwork for credibility is initiated with our expertise in the field [5,28]. The purpose of content analysis is to organise the data collected and to draw a robust investigation of the prominent authors, institutions, sample countries, indicators, and methodologies applied for measuring public sector performance at the macroeconomic level, not the organisational one.

Afonso et al. examined both performance and efficiency of the public sector in 23 industrialised countries by developing a composite index and considering seven sub-indicators divided into two categories, as follows: first category (opportunity indicators) – account administrative, public infrastructure, health, and education, and, on the other hand, the second category ("Musgravian" tasks for the government) – allocation, stabilisation, and distribution, and proxies indicators – socio-economic indicators [29]. To compute the considered indicators, the authors applied a non-parametric framework, namely FDH analysis, with the production possibility frontier. Therefore, the results suggest moderate differences in the public sector across the sample countries. Unsurprisingly, the authors noted that equal income distribution is observed in countries with a large public sector. Furthermore, the results indicate, on the one hand, a significant performance in terms of education, human capital, and income distribution. On the other hand, relatively low performance and economic stability over the EU MS. At the same time, the most significant economic performance is registered in countries with a small public sector.

   

**Table 2. Description of the studies.**

| Authors | Countries | Indicators | Methodologies |
|---|---|---|---|
| Afonso, Schuknecht, and Tanzi for European Central Bank; 2003 | 23 industrialised OECD countries | Composite indicators – Total public sector performance indicator (i) Opportunity indicators: administrative, education, health, and public infrastructure; (ii) Musgravian indicators: income distribution, achievement of stabilisation through economic stability indicator, and economic performance for assessing the allocative efficiency; iii) Proxies: socio-economic indicators. | Non-parametric – FDH analysis production possibility frontier |
| Afonso, Romero, and Monsalve for Inter-American Development Bank; 2006 | Ten "new" EU MS (Member States) Two candidate countries Three "old" member countries (emerging markets) | Composite indicators -performance indicators: administrative, income distribution, health, economic performance and stability, and health; -Musgravian indicators. | Non-parametric – Data Envelopment Analysis (DEA) - production frontier estimation |
| Afonso and Fernandez; 2006 | Lisbon region (51 Portuguese municipalities) | Composite municipal output indicator -total municipal output indicator (TMOI) -process, performance, and effect indicators | Non-parametric – Data Envelopment Analysis (DEA) |
| Angelopoulos, Philippopoulos, and Tsionas; 2008 | 64 developed and developing countries | The composite indicator of public sector efficiency | Non-parametric method – FDH analysis Stochastic production frontier for the public sector 2SLS Panel OLS |
| Afonso, Romero, and Monsalve for Inter-American Development Bank; 2013 | 30 Latin American and 3 Caribbean states | Composite indicators (i) Opportunity indicators: public administration, providers of public services (health, education, and infrastructure); (ii) Indicators by Musgrave: GINI coefficient, distribution, economic, performance, and stability indicators. iii) Proxies: socio-economic indicators. | Non-parametric – DEA (Data Envelopment Analysis) and FSH (Full Disposal Hull) |
| Rouag and Stejksal; 2017 | MENA and Asian NIC countries | Composite indicators and sub-indicators -opportunity indicators: administration, health, education, and infrastructure) and indicators by Musgrave (stability – inflation and GDP growth; distribution – Gini coefficient; and economic performance – GDP real growth and unemployment). | Non-parametric – DEA (Data Envelopment |
| Antonelli and De Bonis; 2016 | 19 European Union countries | Composite performance indicator -implying the indicators of 8 sub-sectors: health, unemployment, labour market, redistribution, poverty, old age, family, and poverty. | Non-parametric method (followed the methodology applied by Afonso A, Schuknecht L and Tanzi (2006)) |
| Lobont, Moldovan, Bociu, Chiș, and Brîndescu-Olariu; 2018 | EU "new" and "old" countries | Composite indicators and sub-indicators -administration, distribution, education, infrastructure, health, stability, and economic performance. | Non-parametric – Factor Analysis – PCA (Principal Component Analysis) |

Afonso et al. examined the public sector's performance and efficiency by constructing different indicators [1]. The methodological endeavour was based on two approaches such as (i) DEA and (ii) Tobit analysis. Furthermore, the results indicate a significant performance in education, human capital, and income distribution, while markedly lower performance and economic stability are associated with the EU MS [1].

Afonso and Fernandes analysed the efficiency of local government spending at a cross-country level. The results revealed that the performance could be improved even without the increase in municipal spending [2].

A comparison of performance in certain areas of economic activities influenced by government intervention with the associated expenditures can be drawn. In this vein, Angelopoulos et al. (2008) constructed a series of sub-indices focusing on four policy areas, namely: (i) administration, (ii) stabilisation, (iii) infrastructure, and (iv) education [4].

A few years later, Afonso et al. deepened the contributions in the research field and investigated the public sector's performance and efficiency by employing different opportunity indicators such as public administration and providers of public services (health, education, and infrastructure), along with Musgrave indicators (distribution, economic, performance, and stability indicator) [1]. The authors analysed the performance and efficiency of the public sector in 30 Latin American and 3 Caribbean states from 2001 to 2010. Thus, the authors applied different methodological approaches, namely DEA and FDH. Therefore, the results suggest significant differences between the analysed countries regarding public sector performance and efficiency.

Conversely, in recent years, due to the direct connection with public budgets and the rational use of resources, the interest given to the performance of the public sector has increased considerably. In this sense, Rouag and Stejskal proposed establishing the ranking of MENA regions based on a composite index approach, applying PCA (principal component analysis) [6]. Thus, the authors ranked the countries by considering two main categories of indicators: opportunity and Musgravian. The results pointed out that MENA countries and Asian NIC states are efficient in the public sector, even if there are different problems related to corruption, high inflation, and lack of political freedom.

Therefore, Afonso and Kazemi recommended the application of non-parametric analysis in their study [11]. Within the same lines, Antonelli and De Bonis [30] constructed a composite index of performance to assess the relative performance of 19 European Union countries in various sub-sectors (health, old age, labour market, redistribution, disability, family, and unemployment), respectively the performance referring to the level at which the objectives and the outcomes proposed by the decision-makers are achieved [30]. The results underlined the degree of variability in the performance index across the EU countries.

However, the measurement of public sector performance is a complex and challenging action. In this light, some authors [5] examined the public sector performance, mainly focusing on identifying a key driver between the six subsectors considered in determining the performance by applying a different approach, namely Principal Component Analysis (PCA) in the process of constructing a composite indicator. The results indicate significant differences between the EU's old and new countries. Moreover, the authors affirmed that modernising and transforming public administration can constitute a primary factor in developing a country. Moreover, the interlinkages and impact of digitalisation on public administration credentials were examined, and the main results attested that good governance stimulates a strong integration of digital technologies, thus contributing to higher performance in the public sector [31].

From the analysed studies, we observe the diversity of the methodologies employed, the non-parametric ones being the most used for measuring the public sector performance, thus proving their robustness.

Although Principal Component Analysis (PCA) and Factor Analysis (FA) are generally applied for the construction of composite indicators when variables are used as explanatory or independent variables in linear regression analysis, Partial Least Squares Regression – PLS represents an innovative and complex alternative because it generates composite variables by considering the response or dependent variable as well, respectively the variables have higher correlations with the response than the composites from their PCA and FA counterparts.

Recent global crises, such as the COVID-19 pandemic, have highlighted even more urgency for developing governance structures, as public administrations around the world have been forced to adapt to new challenges in service delivery, transparency and resource allocation. As digital transformation continues to reshape sectors globally, there is growing recognition of the need for public institutions to leverage technological advances to maintain and improve their performance.

Moreover, specialized literature aims to capture key dimensions of public sector performance, but the changes and challenges in society are complex and multiple, being possible that some studies do not include essential components in the construction of a composite indicator that measures public sector performance, thus leading to an incomplete understanding of the factor that influences public sector performance. In the context of modern challenges, Herasymiuk et al. [32] consider how the digital transformation process and technological innovations affect public governance, considering

digital tools and innovation as important factors in improving the efficiency and transparency of the public governance process, the results of the study highlighting that digital innovations affect the quality and delivery of public services. Alongside, Scupola & Zanfei [33] focused on the co-evolution of public governance and innovation, highlighting that technological and institutional contexts significantly impact the governance of the public sector, with increased attention to the role of public actors and innovation models in the public sector, confirming that the governance of the public sector is strictly linked to the dissemination of policy and more national information locally, while also highlighting the importance of the increasingly high degree of involvement of users, who are becoming increasingly capable of influencing the pace and direction of the innovation process within societies.

Thus, based on the underpinnings, our research paper considers two new domains in the construction of a composite index that measures public sector performance, respectively "Innovation" and "Digitalization", that are not considered by other authors or existing statistics in constructing a composite index that measures the performance of public sector, nuancing the original contribution of our study. Therefore, compared to the existing literature, our study offers different perspectives on measuring the efficiency and performance of the public sector, highlighting the need to consider public governance as a continuous and complex process that can be approached from various perspectives, based on which a series of effective strategies can be developed regarding the implementation and enforcement of digital technology to lead to improving the quality of public service, interaction with citizens, but also to establishing an efficient system, characterized by transparency and effectiveness. Unlike previous studies that constructed a composite indicator that measures the performance of public sector, this study used an analytical approach based on PLS perspective in order to gather evidence. This research paper bridged the gap between the existing theory and practice of EU public sector performance assessment.

Based on the theoretical considerations, the main research hypotheses to be tested have been:

**Hypothesis 1:** Improvements in governance indicators, particularly control of corruption, rule of law, and government effectiveness, positively correlate with overall public sector performance in European countries.

**Hypothesis 2:** The dynamics and characteristics of clusters based on public sector performance have evolved from 2007 to 2021, reflecting shifts in the relative importance of governance, economic, and social factors.

**Hypothesis 3:** The growing importance of the digital economy is reshaping the economic pillar of public sector performance, offering new avenues for enhancing efficiency and service delivery.

**Hypothesis 4:** High-performing clusters, particularly the Nordic and Western European clusters, demonstrate greater resilience to economic and social challenges attributed to robust governance structures and social policies.

## Materials and methods

### Empirical analysis

**Data collection.** This section summarises the statistical approach (i.e., PLS) and model specifications used to investigate public sector performance within the European Union countries. Despite significant advancements in understanding and measuring public sector performance, the existing literature still reveals critical gaps that the study aims to address. Firstly, there is a lack of comprehensive, integrative frameworks combining multiple methodological approaches for assessing public sector performance, particularly those incorporating recent challenges like digital transformation and the impact of global crises such as the COVID-19 pandemic. Most existing studies tend to focus on isolated aspects of public sector performance or employ limited methodological perspectives [5,28,34]. Secondly, there is a notable absence of studies that effectively integrate and analyse the factors affecting public sector performance. These factors include but are not limited to technological advancements, innovation, well-being, and sustainability, alongside traditional metrics like efficiency and governance. A holistic approach that encompasses these diverse domains is essential for a more accurate and nuanced understanding of public sector performance. Thus, the paper focuses on building the composite indicator using 37 indicators extracted from Eurostat (detailed in Appendix A and Appendix B) for

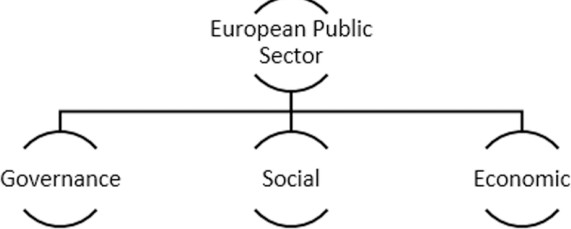

27 countries divided into three pillars of public sector performance: governance, social and economic, as illustrated in Fig 4.

Furthermore, research that applies advanced econometric models, like Partial Least Squares (PLS), is scarce. We consider the PLS approach suitable for the two complementary aims of this research: to build an index of the European Public Sector over the period 2007–2021 and to assess the impact of institutional factors on public sector efficiency playing out across the diverse political and economic landscapes of the EU countries. These models are crucial for handling the data's complex, multidimensional nature and overcoming issues like collinearity among indicators [35–36].

The chosen indicators have attempted to capture various aspects of public sector performance. The justification for their inclusion is based on theoretical foundations, empirical evidence, and policy relevance. Governance plays a fundamental role in shaping public sector performance, as strong institutions lead to better service delivery, economic growth, and public trust. Indicators such as policy implementation effectiveness, government integrity, regulatory quality, law enforcement, and anti-corruption efforts are included because they directly impact institutional strength and democratic stability. Empirical studies have shown that countries with higher institutional quality tend to have more efficient public administrations and resilient economies, justifying their inclusion in the analysis [37; 38]. The digital transformation of public administration is increasingly recognized as a driver of innovation and efficiency. Digital governance indicators—such as e-Government Development Index (EGDI), public sector digitalization, and online service accessibility—serve as proxies for technological innovation in public administration. These indicators are essential in evaluating how digitalization enhances transparency, reduces bureaucratic inefficiencies, and improves citizen engagement [39]. Given the growing role of digital tools in governance, their inclusion reflects modern public sector challenges and advancements. Public sector performance is inherently linked to social outcomes, which reflect the quality of governance and policy effectiveness. Indicators such as Human Development Index (HDI), Quality of Life, Life Expectancy, Education Expenditure, and Healthcare Access Equity are selected because they measure the direct impact of public policies on citizens' well-being. These variables are widely used in global performance assessments and ensure that the study captures both quantitative and qualitative aspects of social progress. Economic stability and growth are both outcomes and determinants of effective public sector governance. Key economic indicators—such as GDP per capita, inflation rate, unemployment rate, and government debt levels—are included because they provide insights into the economic impact of public sector decisions. These indicators allow for assessing the efficiency of fiscal policies, economic resilience, and public sector contributions to economic development. Prior research has demonstrated that well-functioning public administrations positively correlate with sustained economic growth and financial stability. Public investment in infrastructure is a crucial determinant of economic activity and citizen well-being. Indicators measuring the quality of roads, ports, and air transport infrastructure are incorporated to evaluate the extent to which public sector investments contribute to long-term development [40]. These indicators help capture disparities between EU regions and assess whether public infrastructure investments align with economic needs and policy goals.

**Fig 4. Distributed European Public Sector Performance Pillars.**

The countries included in this analysis cover the EU member states over a long period (to 2021 from 2007). The reason for focusing on these countries is that the EU has a common institutional and regulatory framework which enhances some degree of comparability. By including countries at various stages of economic and social development within the EU, the analysis may yield a variety of patterns or constellations of performance across regions. It is also an approach that respects lower public sector efficiency in Eastern as compared to Western EU members – which thus might give valuable policy insights. This combination of indicators and countries provides a comprehensive picture of public sector performance across the EU. It covers governance, social outcomes, digitalization, economic aspects and infrastructure development -all of which are crucial for evaluating and comparing public sector performance in a fast-moving context.

**Data processing.** The paper proposes a robust methodology for handling incompleteness in the public sector data, and imputation is carried out for each country and each indicator. A time series is defined by indicator values for a given country from 2007 to 2021, and for each series, the missing values imputation is performed in the following order: when values are missing at the beginning of (respectively end) of the time series, we duplicate the first (respectively end) of the series; when values are missing in the middle (encircled by available values) of the series, the linear interpolation using years as abscissa and values as ordinates are implied, and for each indicator, the average along with 5th and 95th percentiles are computed. Using the percentiles, the extreme values for each indicator are cut.

LOESS (Locally Weighted Scatterplot Smoothing) non-parametric method [41–42] is applied to each indicator to produce a smoothed curve through the scatterplot of data points and to achieve a robust and optimal smoothing (i.e., ignore outliers). The GDP per capita is used as abscissa in the LOESS smoothing involving the following mathematical steps: select a subset of data points, and weight the data using the tricube weight function, which is defined as $w_i = (1 - |d_i|^3)^3$ where $d_i$ is the normalised distance of the i-th point from the x-value being considered. Then, a simple polynomial regression model is fitted to select and weigh the data. The fitted model is then used to estimate the smoothed value of the y-value (the value of the indicator) corresponding to the given x-value (GDP per capita). This process is repeated for each x-value for each of the 37 indicators. Mathematically, the estimation at each x-value involves solving the weighted least squares problem: $\min \sum_i w_i (y_i - \beta_0 - \beta_1 x_i - \ldots - \beta_k x_k)^2$ where $y_i$ are the values of the indicator, $x_i$ are the values of the GDP per capita, $w_i$ are the weights assigned to each data point and $\beta_0, \beta_1, \ldots, \beta_k$ are the coefficients of the polynomial model. An illustration of the expected value curve for the indicator "Political stability and absence of violence" is illustrated in Fig 5.

**Methodology.** The means of analysis for measuring public sector performance in the European Union consists of two stages: the first is applying Partial Least Squares (PLS) regression to its indicators and second is clustering analysis. PLS regression comprised two main steps.

In the first step, the PLS method has been applied sequentially for the three public performance pillars: governance, economic and social. In the second step, the PLS method addressed all indicators to identify the main determinants of public sector performance.

An important stage has been data normalization and indicator Adjustment. Thus, for each indicator, the minimum value is translated into 1 to avoid division by zero and thus, the indicator value becomes $value = value - \min_{indicator} +1$. A similar approach is also used for the expected value using the following formula $expect\ value = expected\ value - \min_{indicator} +1$ where $\min_{indicator}$ is the minimum of the indicator value for both equations. Then, we calculate the indicator expected value gap using the formula:

$$gap = \frac{value}{expected\ value} - 1$$

The gap (or performance) is multiplied by the indicator sign to define whether the performance is positive or negative. Finally, the weight assigned to an indicator for a given country is obtained by PLS.

                                                                 

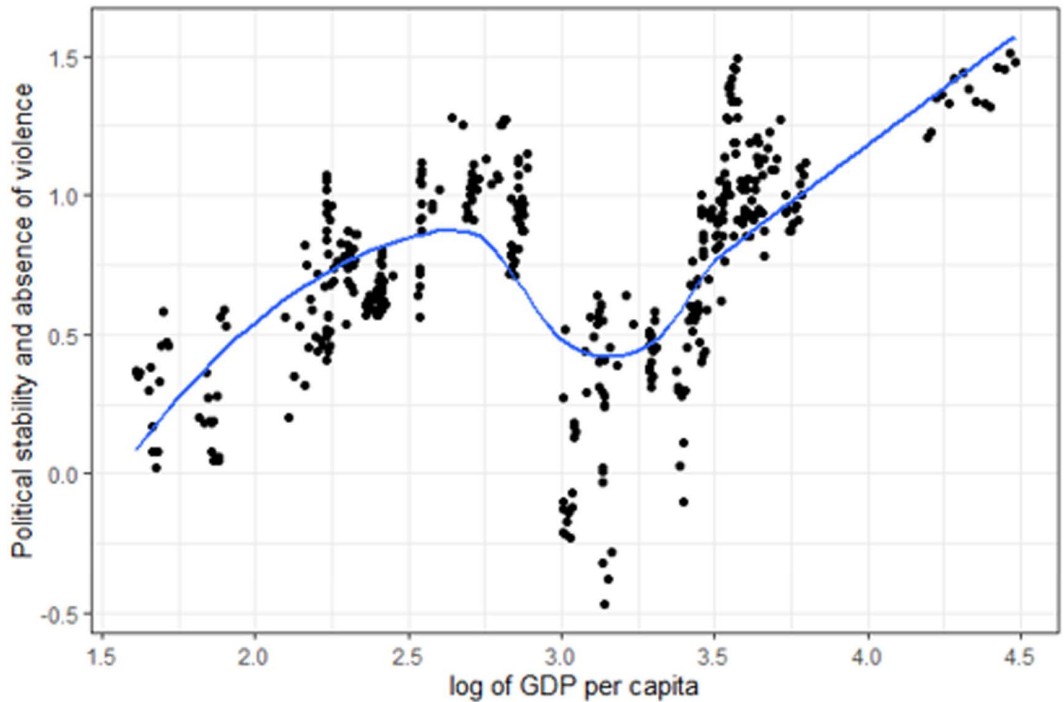

**Fig 5. Value and expected value for the indicator Political stability and absence of violence vs GDP per capita.**

**PLS econometric framework**, developed by Wold [43], allows the construction of predictive models in the presence of many correlated independent variables and comprises three main steps: model formulation, weight vector calculation, indicator weights and performance calculation.

The first step is to formulate the problem by denoting X as the matrix of adjusted indicator values (including gaps and sign adjustments) for all countries and Y as the response matrix representing the outcome of interest (e.g., overall performance score, pillar performance score). PLS aims to find latent variables (components) from X that best predict the response Y by maximising the covariance between projections of X and Y onto these components [44]. The following step computes weight vectors w (for X) and c (for Y) such that covariance between $X_w$ and $Y_c$ is maximised by iteratively extracting actors and deflating X and Y. Then, the component scores t are calculated as $t = Xw$, where w is the weight vector for the first component, and additional components are extracted similarly after deflation. The PLS regression model is then constructed using these component scores to predict Y. The weights of the indicators in the final composite score are derived from the PLS regression coefficients, reflecting the contribution of each indicator to the overall performance measure.

Finally, for each country in each year, the performance is aggregated by pillars using the following formula:

$$Performance_{pillar} = \frac{\sum_{indicator \in pillar} weight_{indicator} * Performance_{indicator}}{\sum_{indicator \in pillar} weight_{indicator}}$$

The final score is obtained by transforming the pillar performance to a scale of 0–100 using the minimum and the maximum performance for the pillar in each year and each country using the formula:

$$Score_{pillar} = 100 * \frac{Performance_{pillar} - min(Performance_{pillar})}{max(Performance_{pillar}) - min(Performance_{pillar})}$$

The second applied method, the cluster analysis approach was used to categorise countries based on their performance, measured by the three pillars of public sector performance: governance, economics, and socials. Cluster analysis, a mainstay in statistical data analysis, enables the discernment of natural groupings in complex datasets, facilitating a nuanced understanding of public sector performance across European Union member states. Determining the optimal number of clusters is a critical preliminary step in cluster analysis. For this purpose, the elbow method, as explained by Ketchen and Shook [45], is employed. This method involves conducting K-means clustering across a range of potential cluster numbers and calculating the total within-cluster sum of squares (WSS) for each. The "elbow point" characterised by a marked change in the rate of decrease of WSS, suggests the most appropriate cluster number. This technique balances the need for cluster homogeneity against the parsimony of the clustering solution [46].

In order to validate the results, the PLS methodology allows the calculation of the variance explained by the dependent variable in the model, which is preferably as close as possible to 100%, but also $Q^2$, a measure specific to the PLS regression that helps to evaluate the model's ability to generalise when encounters new input data by cross validating the data. Thus, a value of $Q^2$ above 0.5 indicates a good prediction model. Finally, four sets of indicators were obtained for the governance, economic, social pillars and over European public.

Building on the insights gained, K-means clustering, as described by MacQueen [47], is implemented to finalise country groupings. This iterative algorithm divides the dataset into K distinct, non-overlapping clusters, minimising the within-cluster sum of squares. This approach aligns with the methodology employed by Steinley [48], ensuring the robustness of the cluster solutions. The algorithm assigns data points to clusters to minimise the squared Euclidean distance to the cluster centroid, a method proven effective in similar studies like those of Hartigan and Wong [49].

The methodology also relies on a robustness analysis built based on the hierarchical clustering executed, following the method delineated by Ward [50]. This agglomerative approach incrementally builds cluster hierarchies, enabling the detailed examination of data structures. A dendrogram is constructed to visually represent the clustering process, offering insights into the natural groupings of countries. The use of Ward's method, which minimises within-cluster variance, aligns with similar applications in other studies [51–52], thereby ensuring analytical rigour.

The integration of these methodologies enables a comprehensive and empirically robust cluster analysis, providing a deep understanding of public sector performance patterns across the European Union. The combined application of the elbow method, hierarchical clustering, and K-means clustering ensures the study's findings are well-grounded in established analytical practices, contributing significantly to public sector performance analysis discourse. The insights derived from this analysis are invaluable for policymakers and stakeholders in benchmarking performance and formulating targeted strategies for public sector improvement.

## Results

Our research examines the dynamics of public sector performance across European Union (EU) countries, introducing the European Public Sector Performance Index, a novel approach that employs Partial Least Squares (PLS) econometric modelling and cluster analysis. The importance of our analysis is highlighted by the results that reveal the dynamics of public sector performance across European Union (EU) countries. The research results capture the multifaceted nature of public sector efficiency by assessing performance across governance, social, and economic dimensions.

**Innovative approach to measure the performance of the European public sector through composite indicators**

**The key drivers of the European public sector considering the governance pillar.** Fig 6 presents the results from a Partial Least Squares (PLS) model for the governance pillar of the European public sector, where the influence of 9 variables on governance is examined. The first five determinants collectively explain 90.14% of the variance within the model. Despite this concentration, the composite indicator of the governance pillar within the European public sector integrates all variables to provide a comprehensive assessment.

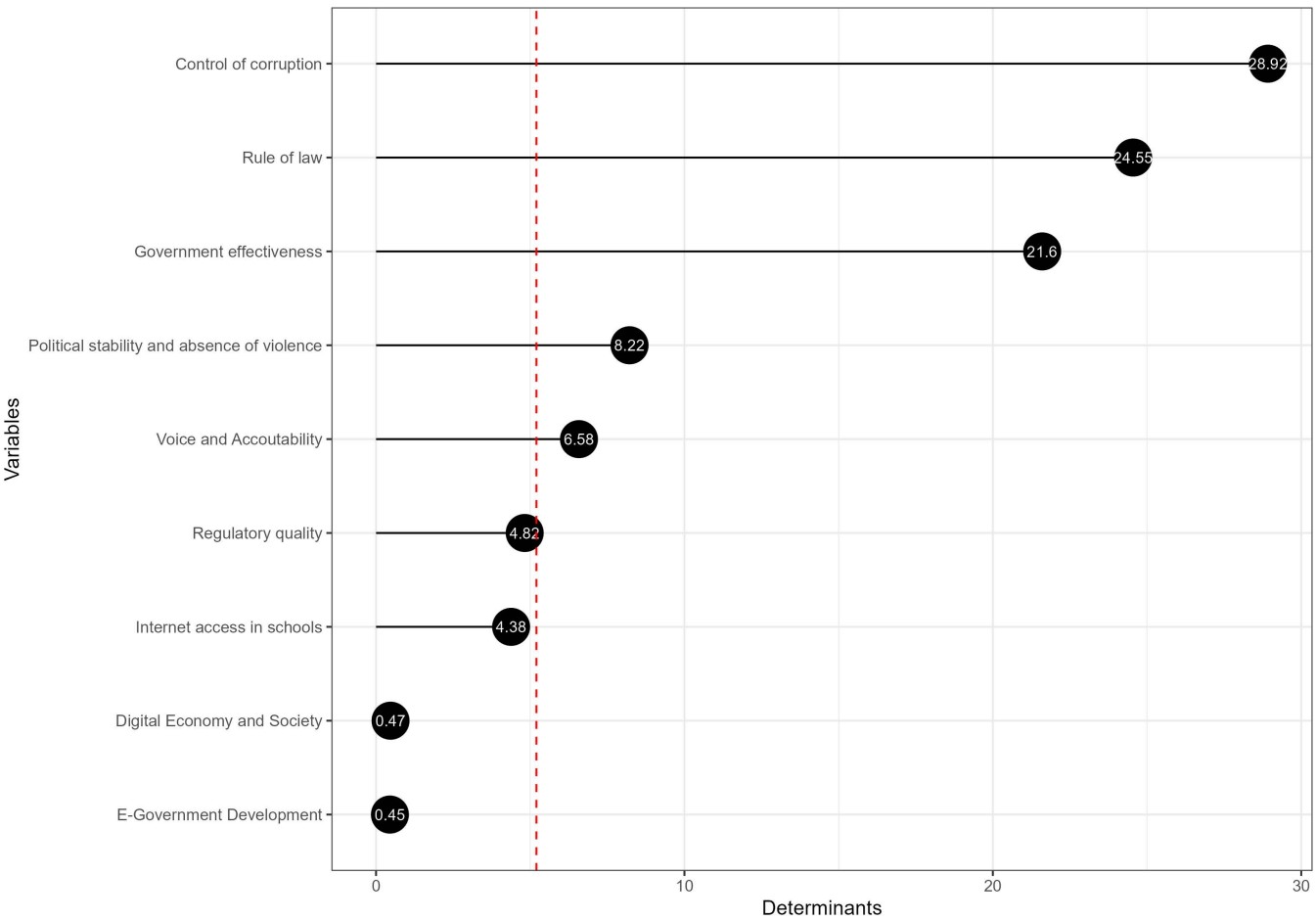

**Fig 6. The main determinants of the governance pillar.**

The empirical results of the governance pillar delineate a hierarchy of determinants critical to the European public sector, each contributing to the multifaceted goal of a balanced and resilient society. The determinant with the highest value is "Control of Corruption", implying that reducing corruption is potentially the most significant driver for improving governance within the European public sector. The important role that the control of corruption plays in governance is highlighted in various studies. Salihu [53] has contributed to the body of literature examining the consequences of corruption, particularly on good governance, identifying the multiple ways corruption has hindered elements of governance. Specific governance dimensions can lead to a high control of corruption, supporting the idea of a hierarchy of determinants in governance-related outcomes [54].

"Rule of Law" and "Government Effectiveness" follow, suggesting a solid legal framework and effective government operations are crucial for governance. Research supports that a robust legal framework and effective government operations are essential for governance. Some authors have found that the rule of law positively impacts government effectiveness in the long run [55]. Additionally, other authors demonstrate that effective corruption control can improve government effectiveness [56]. Their research indicates that democratic regimes and the implementation of robust anti-corruption measures are positively correlated with the effectiveness of governance in the public sector. These studies align with the idea that adherence to the rule of law and the efficient operation of government institutions support good governance.

"Political Stability and Absence of Violence" is subsequent, indicating its importance, albeit less than the top drivers. Research has identified political stability and the absence of violence as influential factors in economic performance and

government effectiveness. For instance, studies have discussed the interplay between social movements, political stability, and their subsequent effects on the economy, suggesting that political conditions and social stability can heavily influence economic performance [57–59]. "Voice and Accountability" follows, emphasising the role of democratic processes and civic participation. The concept of "Voice and Accountability" within the public sector underscores the significance of democratic processes and the involvement of citizens in governance. Studies exploring this include examining the impact of management accounting tools on public sector accountability and transparency, emphasising the importance of quality accounting information for effective management in regional governments [60].

Additionally, while the digital economy and e-government have the most negligible impact presently, they represent areas with potential for future development and could be critical in the long-term evolution of governance.

Despite the varying degrees of influence, the model integrates all variables, highlighting governance's complexity and multifaceted nature. The composite indicator of governance ensures a holistic evaluation, incorporating aspects ranging from technological infrastructure to the rule of law.

**The key drivers of the European public sector considering the economic pillar.** Fig 7 illustrates the economic pillar determinants derived from the PLS model, where 13 variables were examined. The first four determinants explain 79.03% of the variance within the model, and thus, the most influential determinant is "Inflation," with a value surpassing

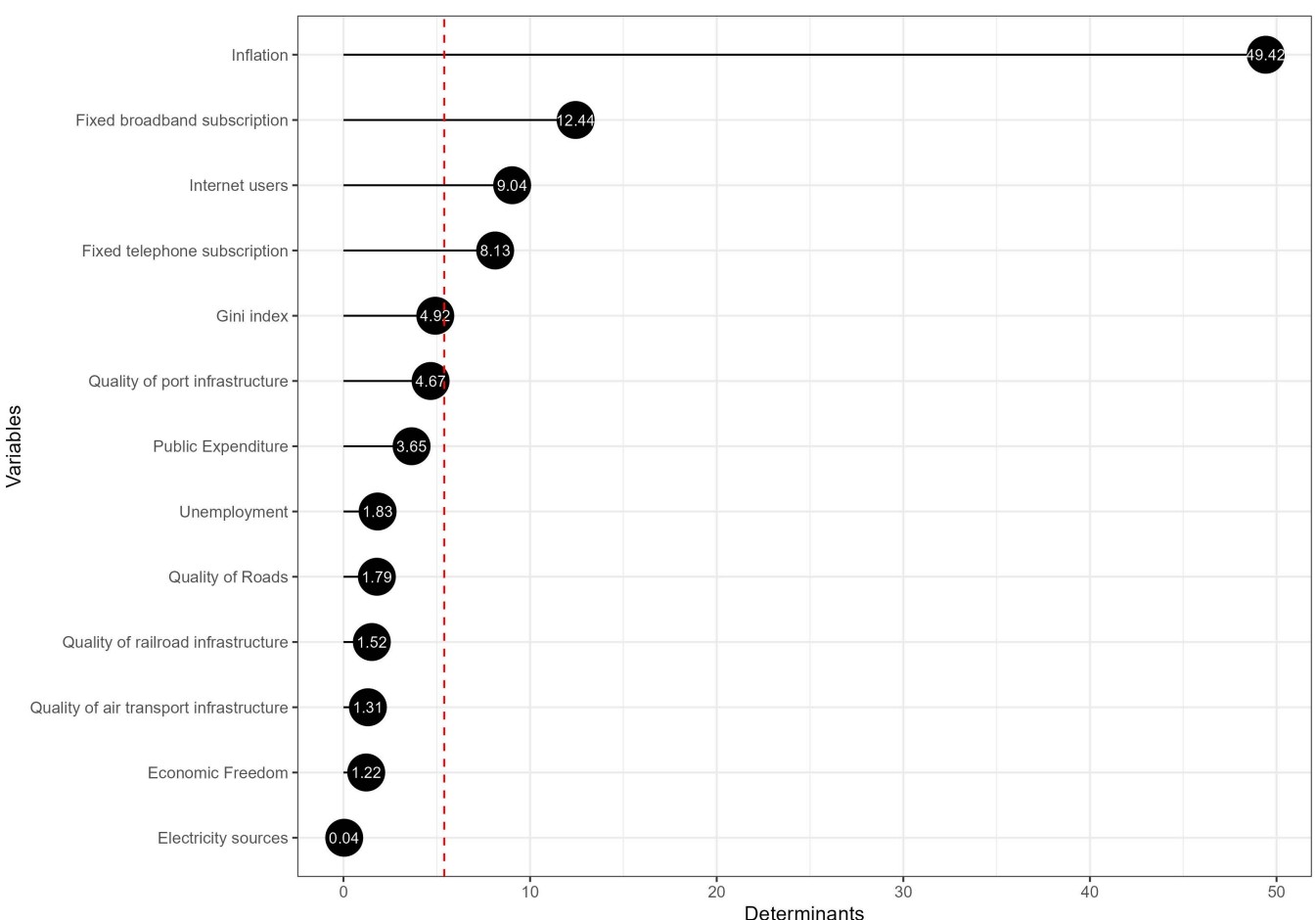

**Fig 7. The main determinants of the economic pillar.**

49.42%, suggesting that price stability is paramount for economic health in the European public sector. The significance of inflation in the context of economic health in the public sector is well-documented in the literature. Inflation, a persistent rise in the general price level, is a key determinant of economic growth. The relationship between inflation and growth has been extensively explored, with a general consensus that while moderate inflation can spur economic activity by encouraging spending and investment, high inflation can undermine economic growth by creating uncertainty and eroding purchasing power [61]. Furthermore, the impact of inflation on economic growth and public finances has been a focal point of empirical studies. The National Bureau of Economic Research (NBER) has investigated the drivers of inflation during the pandemic era, quantifying how domestic inflation was affected by various shocks in multiple countries. The research identifies negative supply shocks and positive aggregate demand shocks due to stimulative policies as essential amplifiers of inflation during the 2020–2023 period [62].

The following determinants, "Fixed broadband subscription" and "Internet users," with values around 12.44% and 9.04%, respectively, underscore the importance of digital infrastructure and connectivity in economic performance. Studies have shown that digitalisation within the public sector can improve social policy and welfare, efficiency in service delivery, and public administration management. A systematic literature review on the digitalisation of the public sector has highlighted the advantages of implementing digital government strategies and the conditions determining policy success or failure, emphasising the significance of embracing digitalisation for public sector innovation and effectiveness [63]. Furthermore, the United Nations Development Programme (UNDP) has conducted a study estimating the economic and human impact of Digital Public Infrastructure (DPI) across various sectors in numerous countries. The study discusses the potential benefits of DPI implementation and how uniform solutions and standards can materialise positive outcomes for people and the planet, showcasing the pivotal role of digital infrastructure in socioeconomic development [64].

On the other hand, variables like "Electricity sources" and "Economic Freedom" have minimal impact based on this model, suggesting that other factors are currently more influential in shaping European economic governance outcomes. However, as the digital economy evolves, these factors may gain importance over time, so they were kept in building the composite indicator.

**The key drivers of the European public sector considering the social pillar.** Fig 8 highlights the determinants derived from the Partial Least Squares (PLS) model for the social pillar of the European public sector, considering 15 proxy variables in this context. The first five determinants collectively explain 70.76% of the variance within the model. Despite this concentration, the composite indicator of the social pillar within the European public sector integrates all variables to provide a comprehensive assessment.

Equity of access to health care services is the variable that explains 21.54% of the variance, suggesting that it is the most significant determinant of the social pillar of the public sector composite indicator. A paper confirms the variable's importance because of public policy's role in ensuring equitable access to healthcare and how this contributes to a nation's overall social health and prosperity [65].

The Legatum Prosperity Index and Education Spending are the following determinants of variance explained by the model (15.93% and 14.08%, respectively), indicating that these factors play a crucial role in the social pillar.

Life Expectancy, Mean Years of Schooling, and Fertility Rate have moderate importance, with weights ranging between 8% and 11%. A cross-sectional found that life satisfaction, less depression, sufficient income, better subjective health, and more education were associated with better quality of life across European countries, indicating the impact of social policies on life expectancy and quality of life [66]. Research examining socioeconomic inequalities in health and life expectancies across Western Europe found that higher education levels were associated with longer life in good health before and after retirement age. This suggests that education significantly influences life expectancy and the distribution of health in later life, which are critical components of social score indices [67].

On the other hand, the different variables have less impact. All the variables were kept in the original Social pillar score to build a more robust index.

                                      

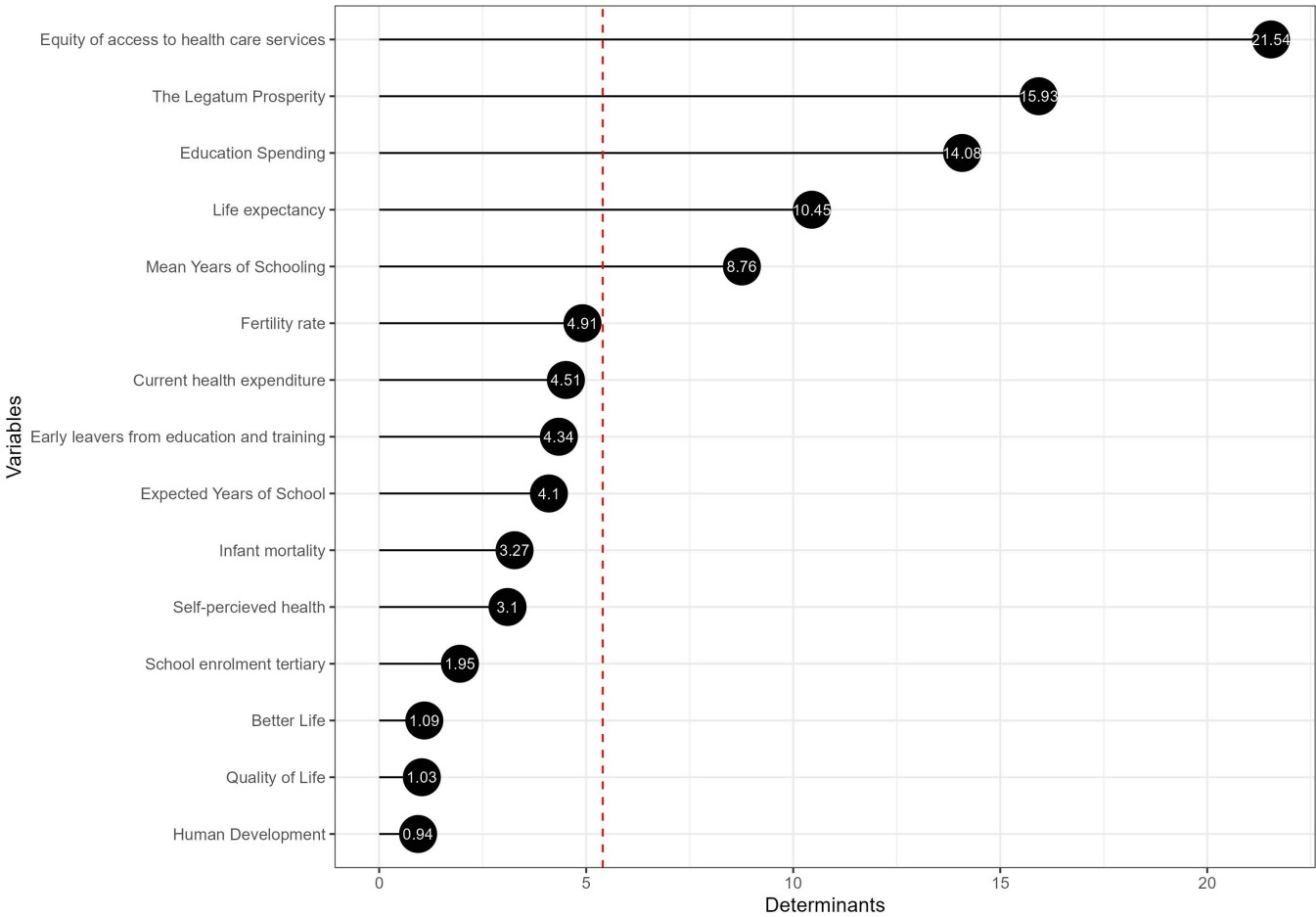

**Fig 8. The main determinants of the social pillar.**

**The key drivers of the overall European public sector.** Fig 9 presents the results from the PLS model for the European public sector index, where the influence of the three pillars' variables is examined. The method allows the introduction of all the variables at once because of its adaptability to reduce multicollinearity. The first five determinants collectively explain 64.56% of the variance within the model. Despite this concentration, the composite indicator of the European public sector integrates all variables to provide a comprehensive assessment. The results will then be compared with the other pillars in terms of countries ranking to check the robustness of the results at the same time.

The strongest determinants for the European public sector index relate to the governance pillar. The prominence of factors like "Control of corruption", "Rule of law", and "Government effectiveness" suggests that transparent and accountable governance structures are perceived as crucial for the public sector's overall performance. This aligns with research suggesting that good governance, characterised by low corruption and effective legal and political systems, is fundamental to economic development, social welfare, and public sector efficiency.

The World Bank often highlights corruption as one of the most important barriers to development. According to the World Bank's World Governance Indicators (WGI), Control of Corruption measures the extent to which public power is exercised for private benefit and assesses the effectiveness of governments in combating corruption. The World Bank identifies corruption as a key constraint to improving governance because it undermines the credibility and legitimacy of institutions and hinders equitable economic development. This variable was chosen for its theoretical relevance, empirical

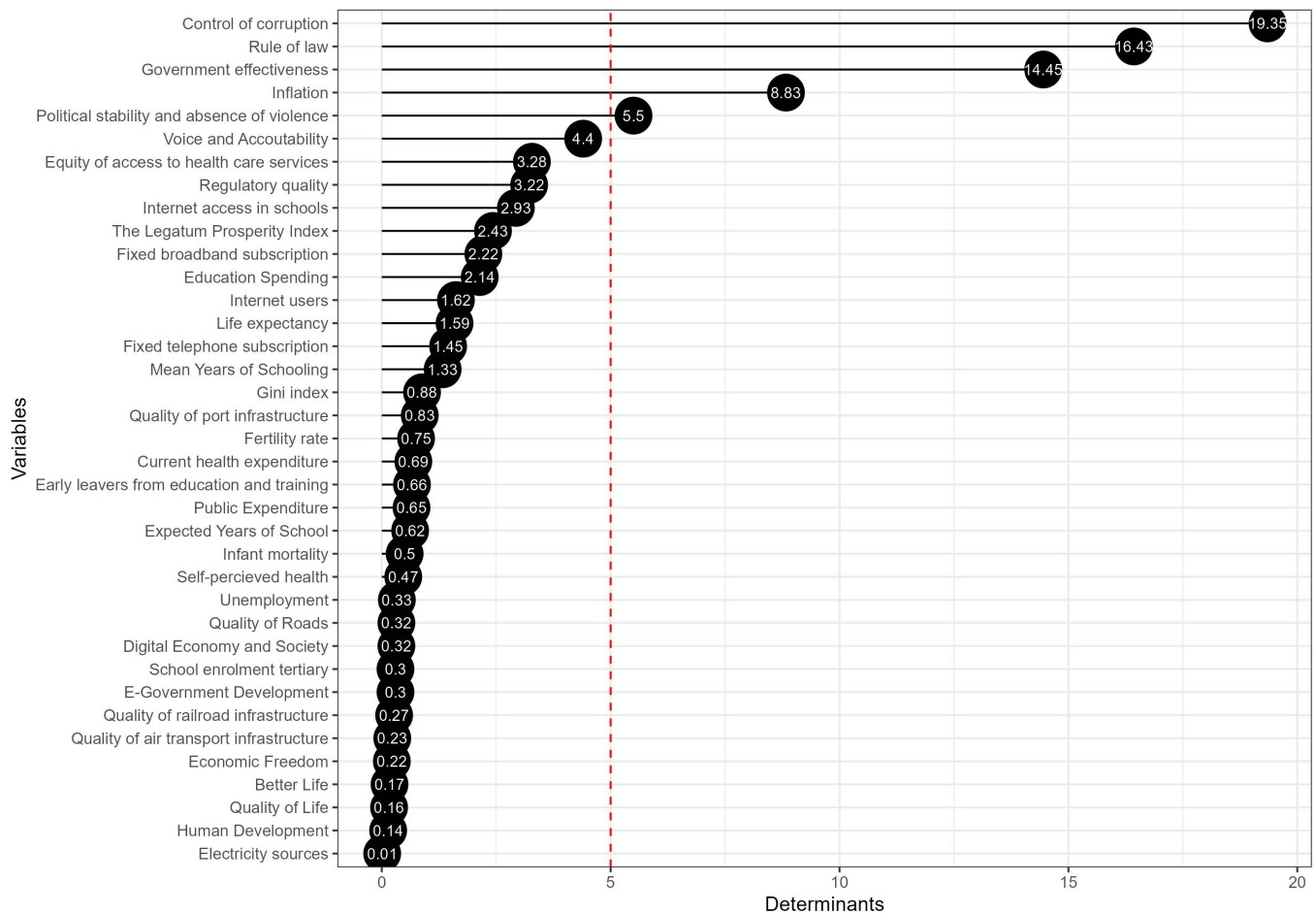

**Fig 9. The main determinants of the overall European Public Sector index.**

support, and alignment with the dimensions of governance that influence public administration performance. Moreover, the World Bank's governance indicators, including Control of Corruption, are widely accepted metrics for assessing, conceptualising and measuring the quality of governance [68–72]."

Although "Inflation" is the only purely economic indicator with a high weight, its significance indicates the importance of economic stability in evaluating the public sector. Inflation affects purchasing power and can indicate economic health or distress. Economists and policymakers often focus on inflation control as a key aspect of macroeconomic policy.

Equity of access to health care services and "Education Spending" are notable social factors, suggesting that social equity and investment in human capital are critical components of the public sector's quality and efficiency.

**Exploration and identification of performance trends and patterns of the European public sector index through cluster analysis.**

**Analysis of the European Public Sector performance through a k-means clustering approach.** Applying the k-means clustering method to the governance, economic and social pillars obtained from PLS for the year 2007, we can identify the main poles of the public sector at the European level in 2007. Dimension 1 on the x-axis explains 76.8% of the variance, which is significant, implying that this dimension is a strong differentiator of public sector performance. Dimension 2 explains a much smaller portion, 18.3%.

Analysing the clusters of countries from Fig 10, we point out the existence of four classes in 2007:

1. Nordic and Western European Cluster (Purple Area): This cluster comprises countries with robust governance structures, advanced social systems, and substantial economic indices. It includes countries such as Austria, Belgium, Denmark, Finland, France, Germany, Ireland, Luxembourg, Netherlands, and Sweden which are typically associated with high standards of public sector performance.

2. Eastern European Cluster (Red Area): Comprising countries such as Estonia, Hungary, Latvia, Lithuania, and Slovenia, this cluster is characterised by transitional economies and evolving governance and social structures, indicative of moderate to developing public sector indices.

3. Central and Eastern European Cluster (Green Area): This grouping, including Croatia, Czechia, Poland, Romania, and Slovakia, suggests a set of countries with possibly comparable levels of public sector development, reflecting shared historical, economic, or political characteristics.

4. Southern European Cluster (Blue Area): Encompassing countries like Bulgaria, Cyprus, Greece, Italy, Malta, Portugal, and Spain, this cluster represents a collection of countries with similar Mediterranean traits, facing everyday challenges and opportunities within their public sectors.

Similarly, the results from 2021 were analysed to determine if the poles of performance in terms of the public sector changed between the first and last periods of the paper. In Fig 11, we point out the existence of 4 classes:

1. Nordic Pinnacle Cluster (Purple Area): This cluster has evolved to predominantly include the Scandinavian countries of Denmark, Finland, and Sweden, which are positioned at the higher end of Dimension 1. Their placement suggests continued excellence and possibly even improvements in governance, social, and economic indices within the public sector.

2. Western European Core Cluster (Red Area): This grouping, with Austria, Belgium, Germany, France, Ireland, Luxembourg, and Netherlands, indicates countries that maintain a stable core position within the European public sector landscape. The shift in their relative positions within this cluster reflects subtle changes or developments in their governance and economic structures.

3. Emerging Eastern European Cluster (Green Area): Countries like Bulgaria, Croatia, Czechia, Italy, Latvia, Lithuania, Poland, Romania, Slovakia, and Slovenia comprise this cluster. The changes in positions, particularly the upward movement along Dimension 2 for Bulgaria and Romania, indicate significant strides in public sector advancement relative to the European context.

4. Mediterranean and Southern European Cluster (Blue Area): This cluster remains consistent with the one from 2007, including countries such as Cyprus, Estonia, Greece, Hungary, Malta, Portugal, and Spain. Their positioning continues to reflect shared characteristics and challenges within their public sectors, with the potential for stability and region-specific dynamics affecting their public sector indices.

When comparing the two results of the two periods, some countries, such as Denmark and Sweden, remain strong performers in the public sector index, with Sweden showing notable improvement. The positions of some countries, like France and the Netherlands, have shifted within their cluster, which may suggest changes in their relative standings.

There is a shift in the importance of dimensions. In Fig 11, Dimension 1 accounts for 67.8% of the variance (a decrease from 2007), and Dimension 2 accounts for 25.3% (an increase from 2007). This suggests that the factors represented by Dimension 2 have become more important in differentiating the public sector performance of these countries.

**A robustness analysis of the European Public Sector performance based on a hierarchical clustering approach.** Applying the hierarchical method of clustering to the three pillars obtained from PLS for 2007, we can identify the main poles of public sector performance at the European level in 2007 (Fig 12).

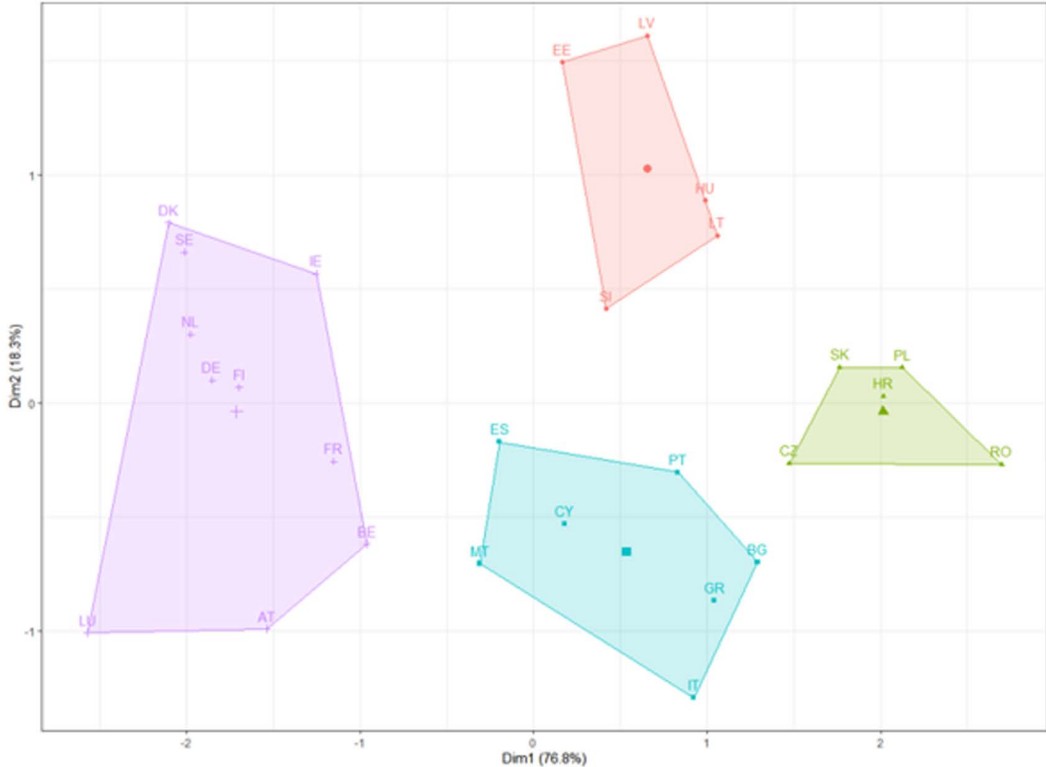

**Fig 10. K-means clustering results for 2007.**

Fig 12 shows the country clusters based on the individual pillar indicators using tree clustering (hierarchical) with centroid clustering and squared Euclidean distances. The similarity between countries in the same cluster decreases as the linkage distance increases. Countries within the same colour-coded cluster have been grouped based on their similarity in terms of the analysed public sector indices. For example, the blue cluster includes countries presumably more similar to each other in public sector performance than those in the red cluster. The greater the height at which two clusters join, the more dissimilar they are. The blue and yellow clusters participate at a lower height than when the grey and red clusters join, indicating that the countries in the blue and yellow clusters have more in common than those in the grey or red clusters.

Analysing the dendrogram of countries, we point out the existence of four classes in 2007:

• Blue Cluster: Cyprus, Spain, Malta, Belgium, France, Austria

• Yellow Cluster: Denmark, Sweden, the Netherlands, Ireland, Finland, Luxembourg, Germany

• Grey Cluster: Estonia, Latvia, Slovenia, Hungary, Lithuania

• Red Cluster: Romania, Croatia, Lithuania, Slovakia, Poland, Bulgaria, Greece, Portugal, Italy.

Analysing the dendrogram of countries, we point out the existence of four classes in 2021 (Fig 13):

• Blue Cluster: Finland, Denmark, Sweden

• Yellow Cluster: Luxembourg, Netherlands, Austria, Belgium, Germany, Ireland, France

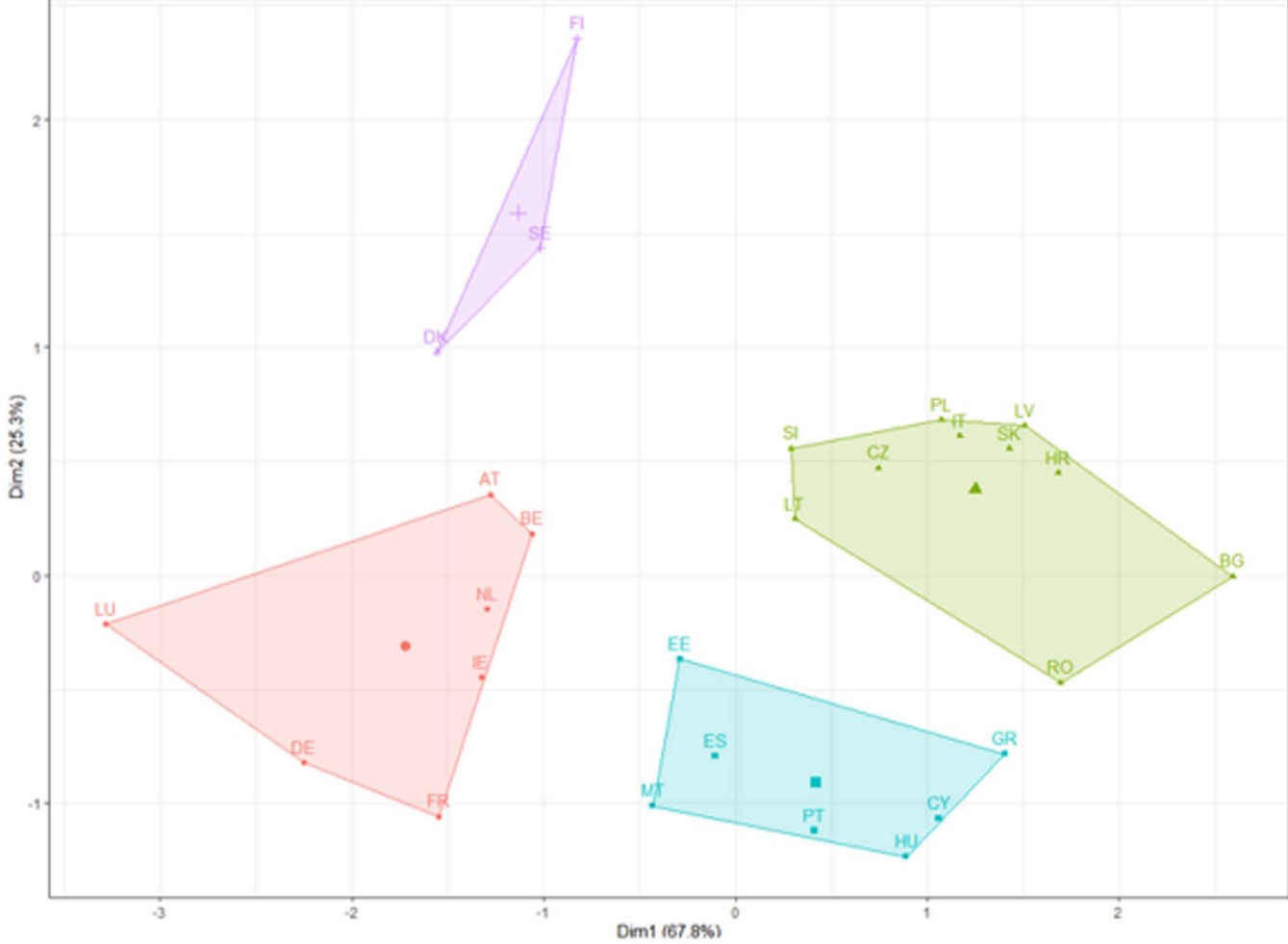

**Fig 11. K-means clustering results for 2021.**

- Grey Cluster: Portugal, Hungary, Cyprus, Greece, Estonia, Spain, Malta

- Red Cluster: Bulgaria, Romania, Slovenia, Lithuania, Latvia, Czech Republic, Slovakia, Croatia, Poland, Italy.

## Discussion

As regards the results from the key drivers of the European public sector considering the governance pillar, Table 3 presents a longitudinal analysis of countries' performance based on the governance pillar from 2007 to 2021. The countries are scored and ranked for each time interval (2007–2011, 2012–2016, 2017–2021, 2020–2021, 2021).

Denmark consistently ranks with perfect or near-perfect scores, indicating strong governance structures. Denmark's governance system has demonstrated agility and resilience, especially when meeting the challenges of digital transformation in the public sector. The country has been recognised for its digital leadership, ranking as one of the most digitised countries in the world. Denmark's governance is characterised by strong state support for development, like that seen in the wind turbine industry, and anticipatory strategies for future challenges, which have put the nation at the forefront of innovative governance practices [73].

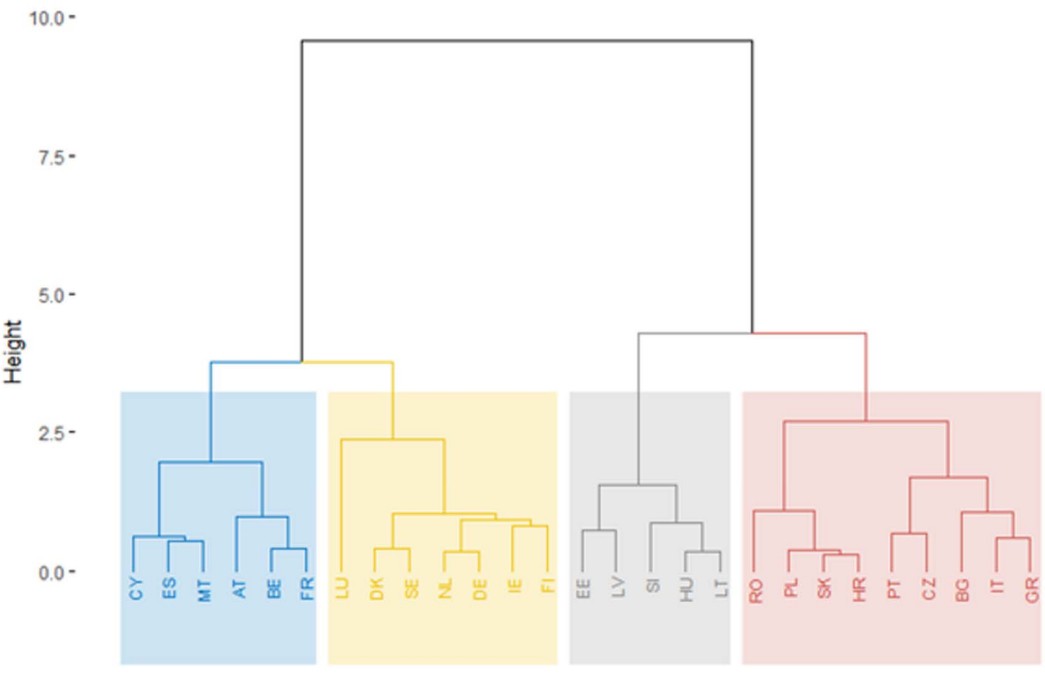

**Fig 12. Public sector performance dendrogram of EU Member States, 2007.**

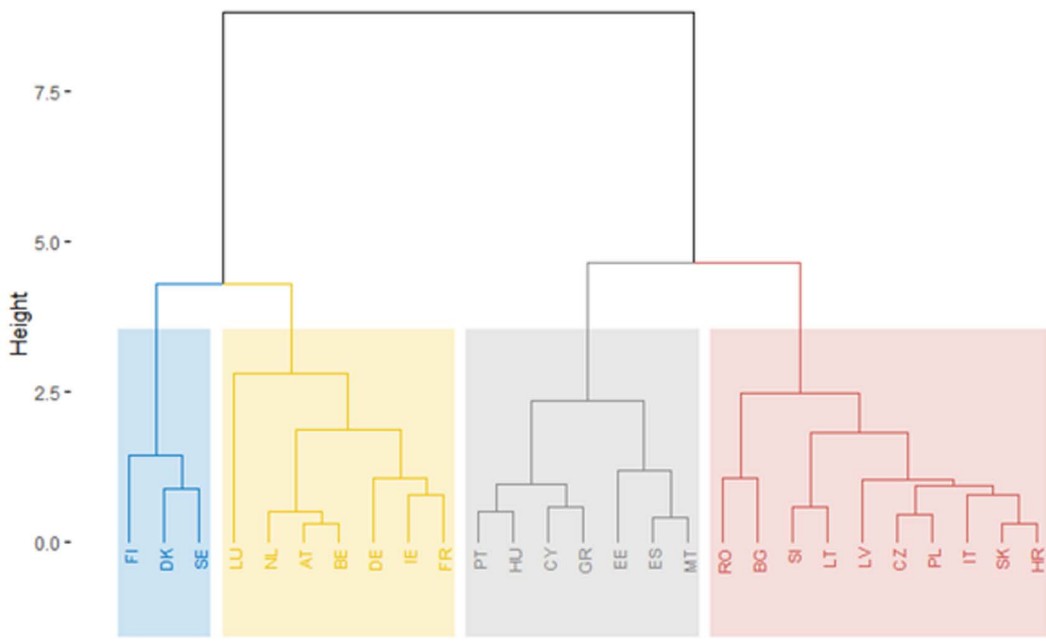

**Fig 13. Public sector performance dendrogram of EU Member States, 2021.**

**Table 3. Comparative analysis of the Governance score (2007–2021).**

| Country | 2007–2011 | | 2012–2016 | | 2017–2021 | | 2020–2021 | | 2021 | |
|---|---|---|---|---|---|---|---|---|---|---|
| | Score | Ranking | Score | Ranking | Score | Ranking | Score | Ranking | Score | Ranking |
| Austria | 78.36 | 6 | 75.50 | 8 | 77.76 | 7 | 76.57 | 7 | 72.77 | 7 |
| Belgium | 68.86 | 10 | 74.12 | 9 | 67.55 | 9 | 66.84 | 10 | 65.88 | 10 |
| Bulgaria | 3.02 | 26 | 0.04 | 27 | 0.19 | 27 | 0 | 27 | 0 | 27 |
| Croatia | 17.78 | 25 | 21.13 | 21 | 22.01 | 23 | 23.87 | 24 | 23.12 | 25 |
| Cyprus | 57.94 | 12 | 52.25 | 13 | 36.91 | 19 | 32.85 | 20 | 31.85 | 20 |
| Czechia | 32.33 | 17 | 36.26 | 19 | 40.05 | 17 | 41.88 | 17 | 42.84 | 16 |
| Denmark | 100 | 1 | 99.95 | 1 | 96.44 | 2 | 98.74 | 2 | 100 | 1 |
| Estonia | 51.30 | 14 | 55.79 | 11 | 65.35 | 11 | 69.99 | 9 | 68.77 | 8 |
| Finland | 94.94 | 2 | 98.25 | 2 | 99.57 | 1 | 98.93 | 1 | 97.86 | 2 |
| France | 70.85 | 9 | 72.48 | 10 | 67.54 | 10 | 64.93 | 11 | 64.83 | 11 |
| Germany | 78.15 | 7 | 83.67 | 6 | 79.76 | 6 | 76.88 | 6 | 75.21 | 6 |
| Greece | 27.13 | 21 | 15.11 | 25 | 15.79 | 25 | 21.72 | 25 | 23.15 | 24 |
| Hungary | 30.53 | 19 | 21.04 | 22 | 23.61 | 21 | 25.65 | 22 | 24.91 | 22 |
| Ireland | 76.91 | 8 | 78.44 | 7 | 70.88 | 8 | 74.77 | 8 | 75.21 | 5 |
| Italy | 23.46 | 24 | 20.51 | 23 | 21.96 | 24 | 25.46 | 23 | 24.65 | 23 |
| Latvia | 29.11 | 20 | 37.67 | 17 | 42.39 | 15 | 46.52 | 14 | 46.61 | 15 |
| Latvia | 31.50 | 18 | 36.38 | 18 | 41.48 | 16 | 42.98 | 16 | 40.77 | 17 |
| Luxembourg | 82.51 | 5 | 86.38 | 5 | 86.46 | 4 | 85.80 | 4 | 81.75 | 4 |
| Malta | 53.90 | 13 | 48.91 | 15 | 39.51 | 18 | 43.18 | 15 | 47.91 | 13 |
| Netherlands | 86.90 | 4 | 88.54 | 4 | 82.74 | 5 | 78.62 | 5 | 66.87 | 9 |
| Poland | 26.60 | 22 | 33.56 | 20 | 30.18 | 20 | 33.54 | 19 | 36.53 | 18 |
| Portugal | 50.47 | 15 | 49.99 | 14 | 46.02 | 14 | 41.58 | 18 | 35.00 | 19 |
| Romania | 0 | 27 | 4.89 | 26 | 7.34 | 26 | 10.77 | 26 | 13.79 | 26 |
| Slovakia | 25.31 | 23 | 18.36 | 24 | 23.26 | 22 | 29.36 | 21 | 30.04 | 21 |
| Slovenia | 48.34 | 16 | 45.32 | 16 | 51.66 | 12 | 55.20 | 12 | 58.95 | 12 |
| Spain | 60.95 | 11 | 54.11 | 12 | 47.97 | 13 | 47.86 | 13 | 46.87 | 14 |
| Sweden | 92.46 | 3 | 93.35 | 3 | 89.35 | 3 | 88.91 | 3 | 87.06 | 3 |

Finland and Sweden also consistently rank high, emphasising the robustness of governance in Nordic countries. This reflects a stable and robust governance system that could contribute to these countries' effective response to crises, including the pandemic. Finland, in particular, has been lauded for its achievements in public sector reform and its focus on continual improvement. Its administration has undergone significant transformations, adopting a participatory approach to governance that prioritises anticipation and systems approaches to complex issues. This aligns with Finland's historical respect for the rule of law and high administrative ethics, contributing to its robust governance and high public trust [74]. Sweden, known for its consensus-based model of public governance, has demonstrated robust welfare state capabilities and a capacity to address horizontal challenges. Shared responsibilities and delegation across different public institutions and levels of government characterise its unique governance system. Swedish governance principles are widely applied within Government Offices, ensuring collective decision-making that aligns with constitutional stipulations [75].

In contrast, countries like Bulgaria and Romania have the lowest rankings, signalling persistent governance challenges that could potentially impact crisis management and public trust. Bulgaria's governance system has faced difficulties in decentralisation and regional development [76], with reforms not achieving the expected outcomes over the last 30 years, resulting in a still largely centralised governance model [77]. Similarly, efforts in Romania have not fully realised the anticipated benefits, with the EU monitoring reforms through mechanisms due to vulnerabilities in governance [78]. Both

countries' paths towards effective governance reforms and digital transformation in the public sector exhibit complexity, indicative of the challenges they face in improving crisis management and public trust [79].

Central European countries like Austria and Germany show strong governance with slight variations in scores, indicating resilience in governance structures [80,81]. Southern European countries, including Greece and Italy, have lower scores and rankings, which may reflect the challenges these countries faced during the pandemic, possibly influencing their governance capabilities [82–83].

Netherlands and Ireland exhibit some variability in their rankings but remain in the upper half of the table, indicating good governance overall [84–85] but with room for enhancements [86–87].

Moreover, based on the results obtained from the analysis regarding the key drivers of the European public sector considering the economic pillar, Table 4 depicts a diverse economic performance across European countries, as measured by several key determinants from 2007 to 2021. Germany shows a consistent upward trajectory, peaking at a perfect score in 2021, underscoring its economic strength and resilience, particularly during and after the pandemic. Germany's regions showed varying resilience during the Great Recession (2007–2017), with larger shares of manufacturing, export orientation, and specialisation associated with more substantial recovery capacities. This indicates a catch-up effect of

**Table 4. Comparative analysis of the Economic pillar score (2007–2021).**

| Country | 2007–2011 | | 2012–2016 | | 2017–2021 | | 2020–2021 | | 2021 | |
|---|---|---|---|---|---|---|---|---|---|---|
| | Score | Ranking | Score | Ranking | Score | Ranking | Score | Ranking | Score | Ranking |
| Austria | 48.54 | 13 | 62.60 | 11 | 57.61 | 13 | 54.74 | 15 | 56.11 | 16 |
| Belgium | 57.39 | 11 | 70.89 | 9 | 63.36 | 12 | 57.27 | 14 | 57.41 | 15 |
| Bulgaria | 16.93 | 24 | 4.95 | 27 | 9.90 | 26 | 8.67 | 26 | 0.00 | 27 |
| Croatia | 18.44 | 22 | 29.55 | 23 | 14.17 | 25 | 7.95 | 27 | 13.60 | 25 |
| Cyprus | 33.07 | 20 | 38.35 | 20 | 37.16 | 20 | 39.81 | 20 | 65.66 | 12 |
| Czechia | 17.42 | 23 | 30.11 | 22 | 28.02 | 22 | 30.51 | 22 | 26.53 | 22 |
| Denmark | 92.14 | 2 | 82.62 | 5 | 85.97 | 6 | 87.52 | 7 | 77.44 | 8 |
| Estonia | 65.28 | 7 | 60.58 | 12 | 57.08 | 14 | 47.76 | 18 | 67.09 | 11 |
| Finland | 65.17 | 8 | 77.30 | 7 | 83.46 | 8 | 88.46 | 6 | 87.52 | 4 |
| France | 79.12 | 6 | 95.99 | 2 | 98.27 | 1 | 92.89 | 3 | 97.78 | 2 |
| Germany | 89.43 | 5 | 94.21 | 3 | 94.65 | 2 | 93.45 | 1 | 100.00 | 1 |
| Greece | 20.62 | 21 | 33.49 | 21 | 47.81 | 17 | 46.09 | 19 | 52.13 | 17 |
| Hungary | 37.75 | 19 | 41.97 | 17 | 54.25 | 15 | 65.67 | 11 | 60.39 | 14 |
| Ireland | 64.79 | 9 | 67.23 | 10 | 71.36 | 9 | 66.43 | 10 | 80.54 | 7 |
| Italy | 44.26 | 14 | 40.11 | 18 | 42.75 | 19 | 48.82 | 17 | 43.98 | 18 |
| Latvia | 52.31 | 12 | 50.98 | 14 | 50.47 | 16 | 58.60 | 13 | 13.59 | 26 |
| Lithuania | 43.55 | 16 | 39.90 | 19 | 32.80 | 21 | 28.26 | 23 | 38.03 | 19 |
| Luxembourg | 95.28 | 1 | 98.60 | 1 | 94.12 | 3 | 92.67 | 4 | 93.45 | 3 |
| Malta | 57.63 | 10 | 74.20 | 8 | 85.80 | 7 | 85.22 | 8 | 81.87 | 6 |
| Netherlands | 89.90 | 4 | 94.18 | 4 | 89.78 | 4 | 90.30 | 5 | 69.53 | 10 |
| Poland | 8.58 | 26 | 8.08 | 25 | 6.52 | 27 | 14.19 | 25 | 15.80 | 23 |
| Portugal | 43.67 | 15 | 44.07 | 16 | 64.87 | 11 | 70.73 | 9 | 64.03 | 13 |
| Romania | 4.17 | 27 | 6.23 | 26 | 24.17 | 23 | 30.75 | 21 | 37.92 | 20 |
| Slovakia | 15.57 | 25 | 20.34 | 24 | 15.99 | 24 | 19.13 | 24 | 14.578 | 24 |
| Slovenia | 40.53 | 17 | 46.69 | 15 | 43.02 | 18 | 49.46 | 16 | 32.28 | 21 |
| Spain | 39.77 | 18 | 51.49 | 13 | 65.50 | 10 | 65.08 | 12 | 72.27 | 9 |
| Sweden | 90.24 | 3 | 82.42 | 6 | 87.08 | 5 | 92.96 | 2 | 85.83 | 5 |

regions with at least average resistance or recovery, highlighting the strengths of regional economic specialisation [88]. An assessment of economic resilience across 52 economies during the early stages of the COVID-19 pandemic ranked Germany 24th, highlighting its solid economic resilience performance compared to many other economies. This study emphasises the utility of an economic resilience index in assessing and enhancing economic resilience under pandemic conditions [89]. Similarly, Luxembourg displays robust economic health, with top rankings most years. Luxembourg's projected growth was bolstered by dynamic domestic demand and growth in the financial sector, demonstrating its economic resilience and capacity to foster exports. Wage growth due to automatic indexation and a slowly declining unemployment rate further underscores the country's robust economic health [90–91].

Denmark, which generally performs well in governance, shows volatility in the economic domain, with a sharp decline in 2021, which might reflect economic challenges or changes in the variables influencing the PLS model. Despite challenges, Denmark's labour market reforms have contributed to a resilient economy [92]. Ireland demonstrates notable improvement over time, suggesting a strengthening economic foundation. Fitzgerald and Honohan [93] discuss the transformation of the Irish economy following its integration into the European Union. This integration shifted Ireland from dependency on agricultural exports to Britain towards becoming a hub for multinational corporations, significantly improving its economic foundation and living standards. This transition also necessitated strengthening education and social policy infrastructures in Ireland.

Southern European countries like Spain and Italy show moderate scores with some improvement over time, but their rankings indicate room for growth compared to their Northern counterparts, as confirmed by scholars [94].

In contrast, countries like Bulgaria, Poland, and Romania consistently rank lower in the rankings. These results suggest that these nations face more significant economic challenges in Europe. Developing the social and solidarity economy in Bulgaria and Romania emphasises these countries' economic and social challenges during their transition. The study discusses the low efficiency of national institutions and social policies and the limited redistributive capacity of the state [95].

Estonia and Latvia present interesting cases with swings in rankings, indicating fluctuating economic conditions or the impact of external factors such as global economic trends or regional policy changes. A study on the Baltic states, including Latvia and Estonia, found that economic growth in these countries significantly depends on general government spending and, thus, the public sector. The development of the Latvian and Estonian economies depends on this spending to a similar extent, with social protection and economic affairs spending in Latvia and social protection and health spending in Estonia explaining a substantial part of economic growth fluctuations [96].

Countries like Sweden and the Netherlands consistently rank high but not at the top, maintaining strong but not leading economic positions in Europe. This suggests high economic development and public sector efficacy, though they face competition from other robust economies [97].

Furthermore, analysing the social pillar scores for European countries from 2007 to 2021 (Table 5) reveals varied social performance trends, which can reflect the broader social policies, economic conditions, and public sector effectiveness within each nation. In terms of top performers, Luxembourg stands out with a perfect score throughout the entire period, maintaining its position at the pinnacle of social achievement. This consistency highlights Luxembourg's exemplary social policies and infrastructure [98–99].

Germany has shown remarkable improvement, particularly in 2021, jumping to a score of 96.83, which places it second. This reflects Germany's solid social systems and ability to adapt and respond effectively to contemporary challenges [100]. Austria consistently scores high, with slight fluctuations, but remains part of the first three performers. Its score in the COVID-19 pandemic emphasises its robust social framework [101]. Sweden, moving up in the rankings to 4th place in 2021 with a score of 80.66, demonstrates its strong commitment to social welfare, illustrating the effectiveness of its social policies [102].

Regarding middle performers, Finland, France, and the Netherlands exhibit strong social scores, indicating their well-established social systems. Their scores reflect a commitment to maintaining high social standards. While the focus

**Table 5. Comparative analysis of the Social pillar score (2007–2021).**

| Country | 2007–2011 | | 2012–2016 | | 2017–2021 | | 2020–2021 | | 2021 | |
|---|---|---|---|---|---|---|---|---|---|---|
| | Score | Ranking | Score | Ranking | Score | Ranking | Score | Ranking | Score | Ranking |
| Austria | 81.13 | 2 | 81.59 | 2 | 72.62 | 3 | 82.58 | 3 | 82.58 | 3 |
| Belgium | 70.72 | 4 | 70.55 | 5 | 65.62 | 5 | 78.75 | 5 | 78.71 | 6 |
| Bulgaria | 19.07 | 18 | 11.60 | 22 | 0.92 | 26 | 0.45 | 26 | 0.50 | 26 |
| Croatia | 15.97 | 23 | 15.41 | 20 | 15.99 | 21 | 19.59 | 21 | 19.61 | 21 |
| Cyprus | 18.83 | 20 | 11.49 | 23 | 0.54 | 27 | 0 | 27 | 0 | 27 |
| Czechia | 26.47 | 15 | 25.18 | 15 | 27.62 | 15 | 34.91 | 15 | 34.93 | 15 |
| Denmark | 59.75 | 9 | 61.16 | 9 | 59.50 | 8 | 70.22 | 9 | 70.21 | 9 |
| Estonia | 12.45 | 24 | 11.80 | 21 | 17.06 | 20 | 25.11 | 18 | 25.10 | 18 |
| Finland | 63.85 | 8 | 64.39 | 8 | 59.46 | 9 | 71.75 | 8 | 71.72 | 8 |
| France | 68.06 | 6 | 68.95 | 6 | 63.11 | 7 | 73.54 | 7 | 73.51 | 7 |
| Germany | 79.42 | 3 | 80.37 | 3 | 80.02 | 2 | 89.12 | 2 | 96.83 | 2 |
| Greece | 18.89 | 19 | 10.99 | 24 | 2.54 | 25 | 3.44 | 25 | 3.44 | 25 |
| Hungary | 7.88 | 25 | 6.03 | 26 | 10.58 | 22 | 14.26 | 22 | 14.30 | 22 |
| Ireland | 46.34 | 12 | 41.32 | 12 | 46.02 | 10 | 62.25 | 10 | 62.20 | 10 |
| Italy | 49.35 | 11 | 47.80 | 11 | 39.88 | 12 | 45.32 | 12 | 45.30 | 12 |
| Latvia | 1.00 | 27 | 0.00 | 27 | 3.01 | 24 | 6.60 | 24 | 6.62 | 24 |
| Lithuania | 22.52 | 17 | 21.33 | 17 | 30.29 | 14 | 44.40 | 13 | 44.39 | 13 |
| Luxembourg | 100 | 1 | 100 | 1 | 100 | 1 | 100 | 1 | 100 | 1 |
| Malta | 49.71 | 10 | 48.64 | 10 | 40.85 | 11 | 47.47 | 11 | 47.44 | 11 |
| Netherlands | 69.07 | 5 | 68.83 | 7 | 65.67 | 4 | 80.37 | 4 | 80.32 | 5 |
| Poland | 16.72 | 22 | 16.33 | 18 | 23.94 | 17 | 33.83 | 17 | 33.82 | 17 |
| Portugal | 27.24 | 14 | 25.47 | 14 | 21.20 | 18 | 24.57 | 19 | 24.52 | 19 |
| Romania | 7.55 | 26 | 6.40 | 25 | 8.90 | 23 | 11.11 | 23 | 11.11 | 23 |
| Slovakia | 16.91 | 21 | 15.88 | 19 | 18.39 | 19 | 23.90 | 20 | 23.90 | 20 |
| Slovenia | 25.54 | 16 | 21.99 | 16 | 24.76 | 16 | 34.42 | 16 | 34.42 | 16 |
| Spain | 39.80 | 13 | 36.64 | 13 | 33.16 | 13 | 38.93 | 14 | 38.89 | 14 |
| Sweden | 66.98 | 7 | 71.22 | 4 | 65.10 | 6 | 74.93 | 6 | 80.66 | 4 |

is on pension systems, the principles of comprehensive social protection, active engagement of social partners, and commitment to fiscal sustainability are integral to these countries' broader social welfare frameworks [103]. Ireland and Italy have shown noteworthy improvements, suggesting strides in enhancing their social infrastructure [104]. Ireland, in particular, has made significant progress, jumping to 10th place in 2021. Spain and Lithuania demonstrate moderate performance, with Spain showing a steady state in its social scores and Lithuania exhibiting a positive trajectory, moving up to the 13th spot by 2021 [105].

Bulgaria and Cyprus are at the lower end of the spectrum, with Cyprus dropping to a score of 0 by 2021, indicating severe challenges in social infrastructure and policy effectiveness [106]. Greece and Hungary, while slightly better than the bottom performers, still face substantial hurdles in achieving higher social standards [107–108]. Estonia, Croatia, Romania and Poland show some progress but remain in the lower tiers, suggesting ongoing challenges in elevating their social welfare systems [109].

Therefore, the main findings from the analysis of the key drivers of the overall European public sector are highlighted in Table 6, which provides the overall European Public Sector index scores from 2007 to 2021, discerning the performance of countries within the three pillars—economic, governance, and social—contributing to the comprehensive public sector performance. Regarding top performers, Luxembourg continues to demonstrate exceptional performance, maintaining a

**Table 6. Comparative analysis of the European Public Sector Index (2007–2021).**

| Country | 2007–2011 | | 2012–2016 | | 2017–2021 | | 2020–2021 | | 2021 | |
|---|---|---|---|---|---|---|---|---|---|---|
| | Score | Ranking | Score | Ranking | Score | Ranking | Score | Ranking | Score | Ranking |
| Austria | 75.25 | 2 | 68.34 | 2 | 60.78 | 3 | 55.24 | 3 | 55.29 | 3 |
| Belgium | 63.94 | 4 | 58.58 | 5 | 54.90 | 4 | 52.81 | 5 | 52.80 | 5 |
| Bulgaria | 4.14 | 25 | 1.76 | 26 | 2.78 | 25 | 2.31 | 26 | 2.36 | 26 |
| Cyprus | 3.03 | 26 | 9.28 | 22 | 0.28 | 27 | 0 | 27 | 0 | 27 |
| Czechia | 24.00 | 16 | 21.74 | 14 | 24.09 | 15 | 24.21 | 15 | 24.35 | 15 |
| Germany | 64.28 | 3 | 65.93 | 3 | 66.06 | 2 | 63.99 | 2 | 63.90 | 2 |
| Denmark | 47.14 | 11 | 49.48 | 9 | 49.14 | 9 | 46.45 | 9 | 46.41 | 9 |
| Estonia | 6.36 | 24 | 8.72 | 23 | 13.64 | 21 | 16.47 | 19 | 16.10 | 20 |
| Spain | 42.15 | 13 | 30.47 | 13 | 27.63 | 13 | 26.00 | 14 | 25.89 | 14 |
| Finland | 56.26 | 7 | 53.83 | 8 | 50.42 | 8 | 48.60 | 7 | 48.58 | 7 |
| France | 59.91 | 5 | 56.41 | 6 | 51.84 | 7 | 48.33 | 8 | 48.37 | 8 |
| Greece | 8.13 | 22 | 9.89 | 21 | 2.76 | 26 | 2.93 | 25 | 2.86 | 25 |
| Croatia | 15.74 | 17 | 13.72 | 20 | 14.97 | 20 | 14.76 | 21 | 14.74 | 21 |
| Hungary | 6.60 | 23 | 5.21 | 25 | 9.28 | 22 | 9.74 | 22 | 9.78 | 22 |
| Ireland | 42.65 | 12 | 33.41 | 12 | 37.85 | 10 | 41.18 | 10 | 40.83 | 10 |
| Italy | 53.18 | 8 | 41.97 | 10 | 35.41 | 11 | 32.21 | 11 | 32.14 | 11 |
| Lithuania | 14.04 | 19 | 18.06 | 17 | 26.16 | 14 | 30.61 | 13 | 30.42 | 13 |
| Luxembourg | 100 | 1 | 100 | 1 | 100 | 1 | 100 | 1 | 100 | 1 |
| Latvia | 0 | 27 | 0 | 27 | 3.21 | 24 | 5.44 | 24 | 5.47 | 24 |
| Malta | 50.34 | 10 | 40.18 | 11 | 33.88 | 12 | 31.45 | 12 | 31.45 | 12 |
| Netherlands | 59.61 | 6 | 55.98 | 7 | 54.12 | 6 | 53.32 | 4 | 53.60 | 4 |
| Poland | 10.22 | 20 | 14.92 | 18 | 21.77 | 16 | 24.07 | 16 | 24.00 | 16 |
| Portugal | 30.88 | 14 | 21.17 | 15 | 17.49 | 18 | 16.25 | 20 | 16.38 | 19 |
| Romania | 8.81 | 21 | 7.12 | 24 | 9.04 | 23 | 8.78 | 23 | 8.63 | 23 |
| Sweden | 52.53 | 9 | 58.65 | 4 | 54.32 | 5 | 50.23 | 6 | 50.43 | 6 |
| Slovenia | 25.58 | 15 | 18.56 | 16 | 21.21 | 17 | 23.58 | 17 | 23.47 | 17 |
| Slovakia | 15.59 | 18 | 14.50 | 19 | 16.95 | 19 | 17.30 | 18 | 17.46 | 18 |

perfect score throughout the period. Its consistent ranking at the top indicates a well-rounded and effective public sector across all pillars. Germany has exhibited strong performance, particularly maintaining its second-place position in recent years. This suggests a robust public sector that has been resilient and adaptable to challenges.

Austria consistently holds a high position, although with some decline in score over the years, which might indicate areas within the economic or social spheres that could be improved. Sweden and the Netherlands also show strong performance, with the Netherlands experiencing a slight increase in scores in the latest year. Both countries have reliable and efficient public sectors with room for targeted improvements. These countries are considered high performers.

Regarding mid-tier performers, Finland and France are among the countries that show a strong public sector with some decline in scores, possibly reflecting the need for policy adaptation in response to new challenges, such as the COVID-19 pandemic or the latest conflicts. Ireland and Italy have improved over time, increasing rankings and scores, indicating progress in strengthening their public sectors.

Bulgaria and Cyprus stand out for their significantly lower scores and rankings. For Cyprus, the drop to zero in the most recent year is particularly concerning and points towards severe challenges in public sector effectiveness. Greece and Hungary remain at the lower end of the spectrum, though not at the bottom, suggesting they face significant hurdles in public sector performance and could benefit from comprehensive reforms.

Romania, Estonia, Croatia, and Poland have shown positive progress over the years, which suggests that while they have faced challenges in their public sectors, there is ongoing development and potential for further improvement. Starting from the bottom, Latvia has shown some progress, although it remains among the countries with the lowest scores.

Table 7 presents the accuracy metrics of the PLS regression models across the governance, economic, and social pillars, as well as for the overall European public sector index. The results indicate high explained variance and strong predictive power ($Q^2$) across all models, confirming the robustness of the analytical framework. Among the individual pillars, the governance model exhibits the highest explained variance (90.14%) and $Q^2$ metric (0.86), highlighting the dominant role of governance-related factors (e.g., Control of Corruption, Rule of Law, Government Effectiveness) in determining public sector performance. The economic pillar model (explained variance = 79.03%, $Q^2$ = 0.81) demonstrates that macroeconomic stability factors (e.g., Inflation, Digital Infrastructure, Economic Freedom) significantly shape public sector efficiency. Similarly, the social pillar model (explained variace = 70.76%, $Q^2$ = 0.75) confirms the importance of social policies (e.g., Equity of Healthcare Access, Education Spending, Life Expectancy) in assessing public sector outcomes. For the overall European public sector index, the model explains 64.56% of variance, with a $Q^2$ of 0.72, indicating that while governance, economic, and social factors collectively drive performance, governance remains the strongest predictor. Overall, the validation results confirm that the PLS regression models are robust, reliable, and well-suited for analyzing public sector performance in the European Union.

## Conclusions

This study's exploration of the European public sector's performance intricately combines governance, economic, and social pillars, contrasting with and extending previous research in its comprehensive methodology and breadth of analysis. Unlike many studies that focus on singular aspects of public sector performance, our approach integrates a wide array of determinants, offering a nuanced understanding that echoes the multifaceted nature of public governance. A pivotal finding across the pillars is the paramount importance of governance factors, particularly "Control of Corruption", "Rule of Law", and "Government Effectiveness", which collectively explain a significant variance within the model. This underscores the critical role of transparent and accountable governance structures in enhancing the public sector's overall performance. For instance, the governance pillar's dominance, with "Control of Corruption" as a leading determinant, aligns with the theoretical perspective that corruption control is instrumental in fostering effective governance, a notion supported by the empirical evidence presented by Salihu [53], who elaborates on corruption's detrimental effects on governance quality. This essential finding, corroborated by the strong performance of countries with robust governance structures like Denmark and Sweden, signifies the foundational necessity of integrity and effectiveness in governance mechanisms for achieving economic stability and social equity. Moreover, the significant role of economic indicators, particularly "Inflation", and social factors like "Equity of access to healthcare services" and "Education Spending" highlights the intricate interplay between economic health, social welfare, and governance in shaping the public sector's efficacy. These results provide a comprehensive assessment of the European public sector's performance and illuminate the critical areas for policy intervention, emphasising the need for robust governance, economic stability, and equitable social policies to enhance public sector efficiency and societal well-being across EU member states. This multidimensional perspective underscores the study's strength, providing a holistic view often lacking in the literature.

**Table 7. Validation of PLS regression models in the four scenarios.**

| Model | Explained variance | $Q^2$ |
|---|---|---|
| Governance pillar | 90.14% | 0.86 |
| Economic pillar | 79.03% | 0.81 |
| Social pillar | 70.76% | 0.75 |
| Overall European public sector | 64.56% | 0.72 |

However, the study is not without limitations. While robust, the reliance on secondary data and the PLS method may not capture the full complexity of the public sector's dynamics or the qualitative aspects of governance and societal well-being. Additionally, the study's focus on EU countries might limit the generalizability of the findings to other regions with different governance structures and socio-economic contexts.

An unexpected finding was the relatively lower importance assigned to digital economy factors in the governance pillar, given the current emphasis on digital transformation as a cornerstone of modern governance. This may suggest a lag in the tangible impact of digital initiatives on governance outcomes or a need for more nuanced indicators to capture this impact effectively.

The significant determinants identified within the governance pillar—Control of Corruption, Rule of Law, and Government Effectiveness—underline the foundational role of solid governance in public sector efficiency, **confirming hypothesis H1**. Countries with higher scores in these areas, like Denmark and Sweden, consistently ranked high in overall public sector performance. The importance of the rule of law lies in its ability to maintain sturdy, predictable, and uniformly enforced legal structures. Nation-fliers boasting efficient legal systems tend to witness enhanced performance in their public sectors, as legal predictability cultivates economic prosperity and stability. Moreover, countries exhibiting robust anti-corruption measures typically boast transparent, accountable, and effective public institutions, resulting in the efficient allocation of resources and the delivery of public services. This supports the hypothesis that governance quality is crucial for public sector effectiveness, resonating with studies highlighting the negative impacts of corruption on governance and the positive correlation between the rule of law and government effectiveness.

The k-means clustering for 2007 and 2021 reveals a noticeable shift in how countries group based on public sector performance. Initially, clusters were more regionally defined, with distinct groupings for Nordic, Western European, Eastern European, and Southern European countries. By 2021, the analysis showed a reconfiguration of these clusters, with some Eastern European countries improving their positions, indicating progress in governance, economic reforms, and social policies.

The evolution of clusters reflects significant changes in the relative importance of different pillars of public sector performance, **confirming the hypothesis H2**. Improving governance indicators like control of corruption, the rule of law, and government effectiveness has contributed to repositioning countries within the clusters. Economic stability, reflected through inflation control and digital infrastructure, alongside social equity, notably in healthcare access and education allocation, have become more pronounced determinants of cluster dynamics.

The hierarchical clustering method further corroborates the evolution of clusters, showing how countries have moved between clusters over time based on their improvements or declines in governance, economic stability, and social equity. Aligning public governance dimensions design with the underlying logic of performance pillars is essential for producing efficient outcomes following the attributes of the public sector and societal value systems.

The economic pillar's analysis underscored "Fixed broadband subscription" and "Internet users" as significant determinants, highlighting the impact of digital infrastructure on economic performance, **confirming hypothesis H3**. The progression in digital infrastructure has facilitated better public service delivery, economic stability, and inclusive growth.

The inclusion and analysis of digital economy indicators in the overall assessment of public sector performance suggest an ongoing shift towards digitalisation. Despite their currently lower impact in the governance pillar, the trend indicates these factors are gaining ground, with potential long-term implications for public sector efficiency.

The increasing reliance on digital platforms and technologies for economic activities has contributed to the resilience of the public sector against economic disruptions. The COVID-19 pandemic, for instance, underscored the importance of digital capabilities in maintaining service delivery and economic functions.

Different research studies address detailed analysis of potential factors contributing to the limited impact of digital economy variables on governance [110–112]. This research perspective could include considerations such as the different levels of digital infrastructure and digital literacy in different regions, the complexity of integrating digital tools into

governance systems, and the possible mismatch between digital transformation and political and institutional contexts. Nevertheless, Zhao [113] the fact that in most countries where the digital economy has been integrated, these do not have major and significant effects on governance given that citizens do not actively participate in the digital process, thus there is a low level of electronic participation opportunities, public decision-makers need to focus more on the acceptance of electronic services by citizens and on the highest possible degree of use in public sector governance structures, the main focus needs to remain on correcting the existing digital gaps. Policymakers may therefore need to adopt a more nuanced and context-specific approach when implementing digital initiatives to ensure that digital economy factors can have a more tangible and positive effect on governance structures.

Across both 2007 and 2021, Nordic countries (Denmark, Sweden, Finland) and certain Western European countries (Germany, Netherlands) consistently appeared in the high-performing clusters, *confirming the hypothesis H4*. This consistency is attributed to their robust governance structures with effective decision-making, resource allocation and coordination, high-quality digital infrastructure, and effective social policies with commitment and adaptability to address societal needs, even in turbulent times. These countries demonstrated adaptability and resilience, particularly in recent global challenges such as the COVID-19 pandemic. Their comprehensive healthcare systems, effective crisis management strategies, and advanced digital economies contributed to maintaining public sector performance.

The analysis highlighted the critical role of governance quality and social equity in these countries' resilience. Robust legal frameworks, transparent institutions, equitable access to healthcare, and substantial investment in education underpin their ability to withstand and quickly recover from economic and social shocks.

In addition, the implications of our results highlight the importance of a public sector performance index, both in terms of findings and limitations. Our main findings also sustain some critical paths of significance for measuring public sector performance. Nevertheless, through the existence of a public sector performance index, decision-makers have to be more responsible and communicative with citizens and stakeholders, ensuring that both parties use the resources efficiently. Moreover, the measurement of public sector performance identifies and highlights the areas in which government improvements are needed, thus guiding public policy adjustments and improving the delivery of quality public services to spur customer satisfaction. On the same lines, our main findings emphasise that having an index that measures the public sector performance facilitates the efficient allocation of resources, directing funding and efforts to programs that exert positive impact and multiple benefits within society. Moreover, transparency, public trust and involvement in government processes are stimulated. The public sector performance assessment can be considered a tool for operating decisions and improving public sector management practices, thus allowing comparisons between different sectors or regions and highlighting the exchange and implementation of best practices.

A distinct focus of our study is dedicated to identifying additional relevant indicators that can be included and measured in our analysis, as well as establishing new measurement procedures and criteria to pave the way for future research directions. This initiative is driven by the recognition that the current framework for evaluating public sector performance may benefit from a broader set of indicators that reflect the complexities of modern governance, particularly in relation to emerging issues such as environmental sustainability and climate change. We aim to continuously update our data by adapting our research framework to include new indicators and dimensions as they emerge. This dynamic approach allows our study to remain relevant and responsive to evolving societal and environmental challenges, particularly those related to climate change. Furthermore, we also consider developing strategies to combat climate change effects based on our analysis results. Therefore, we intend to expand the measurement of the public sector performance by proposing the creation of an index to measure the performance of the public sector and to have the role of an observatory at the European level, accompanied by permanent updates and unlimited availability. This multifaceted approach underscores our recognition of the critical intersection between public sector performance, digital transformation, and environmental sustainability.

The limitations of our research primarily stem from the challenges associated with data availability and the scope of the indicators chosen for analysis. These limitations can impact the overall accuracy and comprehensiveness of our findings, which are crucial for understanding the performance of the public sector in relation to digital transformation and climate change. Furthermore, in certain conditions, another limitation of our research may embody the relatively small group of pillars since there are multiple societal changes, thus not considering all the relevant components that may influence the measurement of the public sector performance. The lack of comprehensive data on specific indicators can lead to gaps in our understanding of how various dimensions of public sector performance are interrelated. Each pillar—be it digitalisation, governance, environmental impact, or societal factors—plays a crucial role in shaping overall performance. Insufficient data may obscure important relationships and nuances that could inform effective policy decisions. Another limitation arises from the relatively small group of pillars included in our analysis. While we aimed to capture key dimensions of public sector performance, societal changes are multifaceted, and many relevant components may not have been considered. This narrow scope can lead to an incomplete understanding of the factors influencing public sector performance.

In conclusion, while our study advances the understanding of public sector performance by integrating governance, economic, and social factors, future research should address its limitations through a broader methodological approach and a wider geographic scope. The significant contribution of this research lies in its holistic approach, bridging the gap between isolated studies of governance practices and the tangible outcomes of public sector performance. Answering critical questions about the determinants of public sector performance, the study sheds light on the foundational role of corruption control, legal robustness, and equitable social policies. It illuminated the paramount importance of governance integrity, economic health, and social equity as interconnected pillars supporting the edifice of public sector efficiency. Additionally, the unexpected findings regarding digital economy factors prompt further investigation into how digital transformation influences governance and public sector performance in the European context and beyond.

## Appendix

Appendix A. Data descriptions and their sources.

| Variables | Description | Source |
|---|---|---|
| Political stability and absence of violence | "Political Stability and Absence of Violence/Terrorism measures perceptions of the likelihood of political instability and/or politically motivated violence, including terrorism" | Worldwide Governance Indicators database, World Bank |
| Control of corruption | "Control of Corruption captures perceptions of the extent to which public power is exercised for private gain, including both petty and grand forms of corruption, as well as "capture" of the state by elites and private interests" | Worldwide Governance Indicators database, World Bank |
| Government effectiveness | "Government Effectiveness captures perceptions of the quality of public services, the quality of the civil service and the degree of its independence from political pressures, the quality of policy formulation and implementation, and the credibility of the government's commitment to such policies" | Worldwide Governance Indicators database, World Bank |
| Regulatory quality | "Regulatory quality captures perceptions of the ability of the government to formulate and implement sound policies and regulations that permit and promote private sector development" | Worldwide Governance Indicators database, World Bank |
| Rule of law | "Rule of law captures perceptions of the extent to which agents have confidence in and abide by the rules of society, and in particular the quality of contract enforcement, property rights, the police, and the courts, as well as the likelihood of crime and violence" | Worldwide Governance Indicators database, World Bank |
| Voice and Accountability | "Voice and Accountability captures perceptions of the extent to which a country's citizens are able to participate in selecting their government, as well as freedom of expression, freedom of association, and a free media" | Worldwide Governance Indicators database, World Bank |

| Variables | Description | Source |
|---|---|---|
| Individuals using the internet for interaction with public authorities | "Within the last 12 months before the survey for private purposes. Derived variable on use of eGovernment services. Individuals used at least one of the following services: for obtaining information from public authorities websites, for downloading official forms, for submitting completed forms" | European Commission |
| Digital Economy and Society Index, by Aggregate score | "The Digital Economy and Society Index (DESI) is a composite index that summarises relevant indicators on Europe's digital performance and tracks the evolution of EU Member States, across five main dimensions: Connectivity, Human Capital, Use of Internet, Integration of Digital Technology, Digital Public Services" | Organisation for Economic Co-operation and Development |
| E-Government Development Index | "The E-Government Development Index presents the state of E-Government Development of the United Nations Member States. Along with an assessment of the website development patterns in a country, the E-Government Development index incorporates the access characteristics, such as the infrastructure and educational levels, to reflect how a country is using information technologies to promote access and inclusion of its people. The EGDI is a composite measure of three important dimensions of e-government, namely: provision of online services, telecommunication connectivity and human capacity" | United Nations |
| Internet access in schools | "The rank of public schools with Internet access" | World Bank |
| Human Development Index | "The HDI is an index capturing life expectancy, expected and average years of schooling and gross national income (GNI)per capita and commonly used by the UN as an index measure of development" | United Nations Development Programme. |
| Expected Years of School | "Expected Years Schooling at age 6. The expected years of schooling (EYS), indicates the future level of education of the population. EYS is defined as the number of years of schooling a child of school entrance age can expect to receive, if prevailing patterns of age-specific enrolment rates persist throughout the child's schooling life" | Global Data Lab |
| Mean Years of Schooling | "Mean years schooling of population 25+. The mean years of schooling of adults aged 25+ (MYS), reflects the current situation with regard to education in a society" | Global Data Lab |
| The Legatum Prosperity Index | "The Prosperity Index has been developed as a practical tool to help identify what specific action needs to be taken to contribute to strengthening the pathways from poverty to prosperity and to provide a roadmap as nations encounter increasing economic and political shocks" | The Legatum Centre for Global Prosperity |
| Early leavers from education and training | "The indicator measures the share of the population aged 18 to 24 with at most lower secondary education who were not involved in any education or training during the four weeks preceding the survey. Lower secondary education refers to ISCED (International Standard Classification of Education) 2011 level 0-2 for data from 2014 onwards and to ISCED 1997 level 0-3C short for data up to 2013. Data stem from the EU Labour Force Survey (EU-LFS)" | Eurostat |
| School enrolment tertiary | "Gross enrollment ratio is the ratio of total enrollment, regardless of age, to the population of the age group that officially corresponds to the level of education shown. Tertiary education, whether or not to an advanced research qualification, normally requires, as a minimum condition of admission, the successful completion of education at the secondary level" | World Bank |

| Variables | Description | Source |
|---|---|---|
| Education Spending %GDP | "These indicators present the total expenditure of general government devoted to three different socio-economic functions (according to the Classification of the Functions of Government - COFOG), expressed as a ratio to GDP. The COFOG divisions covered is education" | Eurostat |
| Equity of access to health care services | "Equity of access to health care services is an index of self-declared unmet need for health care services. It is defined as the percentage of people who self-reported an unmet need for medical care (medical examination or treatment) in the previous 12 months for the following three reasons: financial barriers, waiting times or long distances" | European Commission |
| Self-perceived health | "The indicator on Self-perceived health gives the proportion of people who assess their health to be good or very good" | European Commission |
| Life expectancy at birth total population | "Life expectancy at a given age represents the average number of years of life remaining if a group of persons at that age were to experience the mortality rates for a particular year over the course of their remaining life. Life expectancy at birth gauges the age-specific all-cause mortality rates in an area in a given period" | European Commission |
| Total fertility rate | "The indicator Total fertility rate is computed as the mean number of children that would be born alive to a woman during her lifetime if she were to pass through her childbearing years (generally defined as 15-49) conforming to the fertility rates by age of a given year. It is computed by adding the fertility rates by age for women in a given year (the number of women at each age is assumed to be the same, i.e., mortality is assumed to be zero during the child-bearing period)" | European Commission |
| Infant mortality | "The indicator on Infant mortality gives the ratio of the number of deaths of infants per 1,000 live births based on one year data. Infants are defined as younger than one year of age at death (0-364 days). Eurostat population statistics provide yearly data on infant mortality rates at the EU and regional level as well as for many other European countries" | European Commission |
| Current health expenditure (% of GDP) | "Level of current health expenditure expressed as a percentage of GDP. Estimates of current health expenditures include healthcare goods and services consumed during each year. This indicator does not include capital health expenditures such as buildings, machinery, IT and stocks of vaccines for emergency or outbreaks" | World Bank |
| Quality of Roads | "The Road quality indicator is one of the components of the Global Competitiveness Index published annually by the World Economic Forum (WEF). It represents an assessment of the quality of roads in a given country based on data from the WEF Executive Opinion Survey, a long-running and extensive survey tapping the opinions of over 14,000 business leaders in 144 countries. The road quality indicator score is based on only one question. The respondents are asked to rate the roads in their country of operation on a scale from 1 (underdeveloped) to 7 (extensive and efficient by international standards). The individual responses are aggregated to produce a country score" | World Economic Forum |

| Variables | Description | Source |
|---|---|---|
| Quality of railroad infrastructure | "The Quality of Railroad Infrastructure Indicator is one of the components of the Global Competitiveness Index published annually by the World Economic Forum (WEF). It represents an assessment of the quality of the railroad system in a given country based on data from the WEF Executive Opinion Survey, a long-running and extensive survey tapping the opinions of over 14,000 business leaders in 144 countries. The score for railroad infrastructure quality is based on only one question. The respondents are asked to rate the railroads in their country of operation on a scale from 1 (underdeveloped) to 7 (extensive and efficient by international standards). The individual responses are aggregated to produce a country score" | World Economic Forum |
| Quality of port infrastructure | "The Quality of Railroad Infrastructure Indicator is one of the components of the Global Competitiveness Index published annually by the World Economic Forum (WEF). It represents an assessment of the quality of the railroad system in a given country based on data from the WEF Executive Opinion Survey, a long-running and extensive survey tapping the opinions of over 14,000 business leaders in 144 countries. The score for railroad infrastructure quality is based on only one question. The respondents are asked to rate the railroads in their country of operation on a scale from 1 (underdeveloped) to 7 (extensive and efficient by international standards). The individual responses are aggregated to produce a country score" | World Economic Forum |
| Quality of air transport Infrastructure | "The Quality of air transport infrastructure indicator is one of the components of the Global Competitiveness Index published annually by the World Economic Forum (WEF). It represents an assessment of the quality of airports in a given country based on data from the WEF Executive Opinion Survey, a long-running and extensive survey tapping the opinions of over 14,000 business leaders in 144 countries. The score for air transport infrastructure quality is based on only one question. The respondents are asked to rate the passenger air transport in their country of operation on a scale from 1 (underdeveloped) to 7 (extensive and efficient by international standards). The individual responses are aggregated to produce a country score" | World Economic Forum |
| Inflation | "Inflation, as measured by the consumer price index, reflects the annual percentage change in the cost to the average consumer of acquiring a basket of goods and services that may be fixed or changed at specified intervals, such as yearly. The Laspeyres formula is generally used" | World Bank |
| General government gross debt | "The Treaty on the Functioning of the European Union defines this indicator as the ratio of government debt outstanding at the end of the year to gross domestic product at current market prices. For this calculation, government debt is defined as the total consolidated gross debt at nominal value in the following categories of government liabilities (as defined in ESA 2010): currency and deposits (AF.2), debt securities (AF.3) and loans (AF.4). The general government sector comprises the subsectors of the central government, state government, local government and social security funds. Please refer to the Eurostat Manual on Government Deficit and Debt for further methodological guidance and interpretation. Total government gross debt in million EUR is shown as well" | Eurostat |

| Variables | Description | Source |
|---|---|---|
| Public Expenditure | "Total expenditure comprises all transactions recorded under positive uses in the ESA framework, and subsidies payable, in the current accounts as well as transactions (gross capital formation, acquisition fewer disposals of non-financial non-produced assets plus capital transfers payable) in the capital account of the government" | Eurostat |
| Economic Freedom Summary Index | "The index published in Economic Freedom of the World measures the degree to which the policies and institutions of countries are supportive of economic freedom. The cornerstones of economic freedom are personal choice, voluntary exchange, freedom to enter markets and compete, and security of the person and privately owned property. Forty-two data points are used to construct a summary index, along with a Gender Legal Rights Adjustment to measure the extent to which women have the same level of economic freedom as men. The degree of economic freedom is measured in five broad areas." | Cato Institute |
| Unemployment | "The unemployment rate is the number of unemployed persons as a percentage of the labour force (the total number of people employed plus unemployed)" | Eurostat |
| GDP per capita | "GDP per capita is gross domestic product divided by midyear population. GDP is the sum of gross value added by all resident producers in the economy plus any product taxes and minus any subsidies not included in the value of the products. It is calculated without making deductions for the depreciation of fabricated assets or for the depletion and degradation of natural resources. Data are in current U.S. dollars" | World Bank |
| Gini index | "Gini index measures the extent to which the distribution of income (or, in some cases, consumption expenditure) among individuals or households within an economy deviates from a perfectly equal distribution. A Lorenz curve plots the cumulative percentages of total income received against the cumulative number of recipients, starting with the poorest individual or household. The Gini index measures the area between the Lorenz curve and a hypothetical line of absolute equality, expressed as a percentage of the maximum area under the line. Thus a Gini index of 0 represents perfect equality, while an index of 100 implies perfect inequality" | World Bank |

## Appendix B. Descriptive statistics.

| Variable | Min | Median | Mean | St. dev | Max | Skewness | Kurtosis | Jarque-Berra | Prob. |
|---|---|---|---|---|---|---|---|---|---|
| Political stability and absence of violence | −0,47 | 0,78 | 0,74 | 0,37 | 1,51 | −0,49 | 3,17 | 17,02 | 0,00 |
| Control of corruption | −0,50 | 0,87 | 0,98 | 0,79 | 2,45 | 0,19 | 1,80 | 26,61 | 0,00 |
| Government effectiveness | −0,36 | 1,06 | 1,09 | 0,58 | 2,35 | −0,17 | 2,41 | 7,88 | 0,01 |
| Regulatory quality | 0,15 | 1,15 | 1,18 | 0,44 | 2,05 | −0,02 | 1,98 | 17,56 | 0,00 |
| Rule of law | −0,13 | 1,07 | 1,11 | 0,60 | 2,13 | −0,17 | 1,99 | 19,02 | 0,00 |
| Voice and Accountability | 0,26 | 1,07 | 1,09 | 0,34 | 1,69 | −0,36 | 2,39 | 14,95 | 0,00 |
| Individuals using the Internet for interaction with public authorities | 5,00 | 48,00 | 47,70 | 20,65 | 92,25 | 0,13 | 2,30 | 9,16 | 0,01 |
| Digital Economy and Society Index, by Aggregate score | 19,36 | 37,78 | 38,45 | 9,72 | 70,06 | 0,32 | 2,90 | 7,06 | 0,02 |
| E-Government Development Index | 0,54 | 0,75 | 0,75 | 0,10 | 0,99 | −0,06 | 2,14 | 12,76 | 0,00 |

| Variable | Min | Median | Mean | St. dev | Max | Skewness | Kurtosis | Jarque-Berra | Prob. |
|---|---|---|---|---|---|---|---|---|---|
| Internet access in schools | 3,26 | 5,16 | 5,14 | 0,79 | 6,61 | −0,31 | 2,27 | 15,61 | 0,00 |
| Human Development Index | 0,76 | 0,88 | 0,87 | 0,04 | 0,96 | −0,23 | 2,33 | 11,19 | 0,00 |
| Expected Years of School | 13,50 | 16,10 | 16,10 | 1,31 | 18,00 | −0,12 | 1,97 | 18,54 | 0,00 |
| Mean Years of Schooling | 7,70 | 12,10 | 11,84 | 1,18 | 14,20 | −0,11 | 1,99 | 20,05 | 0,00 |
| The Legatum Prosperity Index | 61,02 | 72,91 | 73,50 | 6,16 | 84,27 | 0,01 | 1,91 | 17,21 | 0,00 |
| Early leavers from education and training | 2,20 | 9,30 | 10,44 | 5,42 | 36,50 | 1,67 | 7,07 | 467,85 | 0,00 |
| School enrolment tertiary | 10,61 | 68,51 | 69,02 | 19,29 | 150,88 | 0,37 | 6,51 | 217,84 | 0,00 |
| Education Spending %GDP | 2,80 | 5,10 | 5,07 | 0,94 | 7,20 | 0,00 | 2,38 | 6,43 | 0,03 |
| Equity of access to health care services | 0,00 | 1,90 | 3,22 | 3,48 | 16,40 | 1,72 | 5,61 | 316,13 | 0,00 |
| Self-perceived health | 3,50 | 21,30 | 23,04 | 11,04 | 55,80 | 0,75 | 3,31 | 39,58 | 0,00 |
| Life expectancy at birth total population | 70,70 | 80,60 | 79,36 | 3,01 | 84,00 | −0,72 | 2,31 | 43,28 | 0,00 |
| Total fertility rate | 1,14 | 1,53 | 1,55 | 0,19 | 2,06 | 0,54 | 2,68 | 21,21 | 0,00 |
| Infant mortality | 1,30 | 3,50 | 3,88 | 1,65 | 12,00 | 1,71 | 6,62 | 417,88 | 0,00 |
| Current health expenditure (% of GDP) | 4,70 | 8,14 | 8,22 | 1,80 | 11,70 | 0,09 | 1,86 | 22,22 | 0,00 |
| Quality of Roads | 1,91 | 4,94 | 4,75 | 1,15 | 6,72 | −0,49 | 2,36 | 23,18 | 0,00 |
| Quality of railroad infrastructure | 2,00 | 4,35 | 4,22 | 1,09 | 6,53 | −0,11 | 2,23 | 10,74 | 0,00 |
| Quality of port infrastructure | 2,62 | 4,90 | 4,88 | 0,95 | 6,81 | −0,12 | 2,36 | 7,80 | 0,02 |
| Quality of air transport infrastructure | 3,18 | 5,39 | 5,18 | 0,83 | 6,71 | −0,37 | 2,26 | 18,43 | 0,00 |
| Inflation | −2,10 | 1,73 | 1,97 | 2,08 | 15,40 | 1,88 | 10,28 | 113,10 | 0,00 |
| General government gross debt | 3,80 | 57,10 | 64,71 | 37,60 | 206,30 | 0,98 | 4,06 | 84,58 | 0,00 |
| Public Expenditure | 24,30 | 45,30 | 45,57 | 6,85 | 64,90 | −0,11 | 2,94 | 0,98 | 0,62 |
| Economic Freedom Summary Index | 6,63 | 7,70 | 7,66 | 0,29 | 8,32 | −0,91 | 4,15 | 78,29 | 0,00 |
| Unemployment | 2,00 | 7,40 | 8,53 | 4,41 | 27,50 | 1,68 | 6,28 | 372,94 | 0,00 |
| GDP per capita | 5,00 | 20,58 | 23,62 | 15,34 | 88,21 | 1,51 | 6,35 | 342,64 | 0,00 |
| Gini index | 23,20 | 31,20 | 31,33 | 3,65 | 41,30 | −0,01 | 2,41 | 5,77 | 0,05 |

## Supporting information

**S1 File. European public sector.**
(ZIP)

## Author contributions

**Conceptualization:** Adriana AnaMaria Davidescu, Oana-Ramona Lobonț, Eduard Mihai Manta.

**Data curation:** Eduard Mihai Manta, Lavinia Daniela Mihit, Alexandra Madalina Taran.

**Formal analysis:** Adriana AnaMaria Davidescu, Eduard Mihai Manta, Lavinia Daniela Mihit, Alexandra Madalina Taran.

**Investigation:** Adriana AnaMaria Davidescu, Oana-Ramona Lobonț, Lavinia Daniela Mihit.

**Methodology:** Adriana AnaMaria Davidescu, Eduard Mihai Manta, Alexandra Madalina Taran.

**Software:** Adriana AnaMaria Davidescu, Eduard Mihai Manta, Alexandra Madalina Taran.

**Supervision:** Adriana AnaMaria Davidescu, Oana-Ramona Lobonț.

**Validation:** Adriana AnaMaria Davidescu, Oana-Ramona Lobonţ.

**Visualization:** Oana-Ramona Lobonţ, Lavinia Daniela Mihit.

**Writing – original draft:** Eduard Mihai Manta, Alexandra Madalina Taran.

**Writing – review & editing:** Adriana AnaMaria Davidescu, Oana-Ramona Lobonţ.

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
