## [Decision Letter · Decision Letter 0]

PONE-D-24-38083Bridging gaps in EU public sector performance assessment: An analytical approach based on PLS perspectivePLOS ONE

Dear Dr. Taran,

Thank you for submitting your manuscript to PLOS ONE. After careful consideration, we feel that it has merit but does not fully meet PLOS ONE’s publication criteria as it currently stands. Therefore, we invite you to submit a revised version of the manuscript that addresses the points raised during the review process.

**ACADEMIC EDITOR:** Please revise the paper according to the recommendations of the reviewer. 

We look forward to receiving your revised manuscript.

Kind regards,

Valentina Diana Rusu, PhD

Academic Editor

PLOS ONE

Journal Requirements:

4. We note you have included a table to which you do not refer in the text of your manuscript. Please ensure that you refer to Table 2 in your text; if accepted, production will need this reference to link the reader to the Table.

Reviewers' comments:

Reviewer's Responses to Questions

**Comments to the Author**

1. Is the manuscript technically sound, and do the data support the conclusions?

Reviewer #1: Yes

Reviewer #2: Yes

2. Has the statistical analysis been performed appropriately and rigorously? 

Reviewer #1: Yes

Reviewer #2: Yes

3. Have the authors made all data underlying the findings in their manuscript fully available?

Reviewer #1: Yes

Reviewer #2: Yes

4. Is the manuscript presented in an intelligible fashion and written in standard English?

Reviewer #1: Yes

Reviewer #2: Yes

5. Review Comments to the Author

Reviewer #1: This paper addresses an interesting topic related to the EU public sector performance assessment. I acknowledge the amount of work invested in preparing the manuscript and consider it significantly contributes to the scientific literature. I would only suggest to the author(s) to enhance the policy implications of own findings and limitations / future research directions, both could be augmented.

Reviewer #2: Journal: PLOS ONE

Article title: Bridging gaps in EU public sector performance assessment: An analytical approach

based on PLS perspective

Manuscript ID: PONE-D-24-38083

General Comments:

This article studies the dynamics of public sector performance across European Union (EU) countries through a comprehensive methodological framework. The authors use European Public Sector Performance Index, a novel approach that employs Partial Least Squares (PLS) econometric modelling and cluster analysis to evaluate public sector performance from 2007 to 2021. The authors reached the conclusion that significant determinants of performance, including governance factors like control of corruption, rule of law, and government effectiveness, as well as economic indicators such as Inflation and social factors like equity of access to healthcare services and education spending,.

Overview:

The paper is very well written and the empirical work does appear to be carefully and correctly done. The research question is very good and it does make a sufficient new contribution to the literature to be suitable for the PLOS ONE ONLY after MINOR revisions.

In fact, the literature on the dynamics of public sector performance across European Union (EU) is quite new.

The contribution of the paper is the analysis of European Public Sector Performance Index, a novel approach that employs Partial Least Squares (PLS) econometric modelling and cluster analysis to evaluate public sector performance from 2007 to 2021.

The paper is very interesting, and in my view, it needs to be significantly improved to reach the standard required for publication in this journal.

Specific Comments:

1. Title: quite long; try to reduce to one sentence (can be removed: An analytical approach

based on PLS perspective)

2. Abstract: very large; try to reduce with the present results from the article

3. Introduction: NOVELTY + results (better explanation); very short, increase to maximum to 2 pages

4. Methodology: why the authors use only these indicators into the model and only these countries? Present some theoretical explanations for these indicators

5. Explain the construction of the model used for analysis.

6. Discussions: at least 2 pages; separate from the conclusions

General considerations: the idea of the article is very good, and the construction of the article is sometimes very detailed and technical. The authors MUST improve the methodology and discussions, and change the article accordingly. I ONLY recommend this article be published in PLOS ONE after MINOR revisions (methodology and discussion).

6. PLOS authors have the option to publish the peer review history of their article (what does this mean? ). If published, this will include your full peer review and any attached files.

**Do you want your identity to be public for this peer review?** For information about this choice, including consent withdrawal, please see our Privacy Policy .

Reviewer #1: No

Reviewer #2: No

---

## [Author Response · Author response to Decision Letter 1]

28 Nov 2024

In the attention of:

Emily Chenette

Editor-in-Chief, Journal PLOS ONE

November 8th, 2024

Dear Editor-in-Chief,

Dear Anonymous Reviewers,

We express our gratitude for your reconsideration of the manuscript titled “Bridging Gaps in EU Public Sector Performance Assessment: An Analytical Approach Based on PLS Perspective” with manuscript ID PONE-D-24-38083.

We would like to extend our deepest gratitude to you for your thorough review of our manuscript and for allowing us to revise and enhance the quality of the final version. We are greatly encouraged by your thoughtful feedback, and we are revising our paper with the utmost care and dedication.

In response to the valuable comments provided, we have carefully addressed each point to ensure the overall improvement of the manuscript. Below are our detailed responses to each observation and the corresponding revisions made. Our manuscript has been revised considering the comments from Reviewers 1 and 2, and all changes in the revised manuscript were introduced in red. Additionally, we have made several other revisions to enhance the quality of the paper, including improving the grammar and spelling throughout the manuscript.

We believe that these changes to the manuscript will facilitate the decision to publish this study in the Journal PLOS ONE. We have made a substantial effort to consider the essential suggestions proposed by the reviewers and remain open to consideration of any further comments on our answers.

Sincerely,

Assistant Professor, PhD Alexandra-Mădălina Țăran

JOURNAL REQUIREMENTS

Manuscript ID: PONE-D-24-38083

Journal: PLOS ONE

Article title: Bridging Gaps in EU Public Sector Performance Assessment: An Analytical Approach Based on PLS Perspective

Point 1: Please ensure that your manuscript meets PLOS ONE's style requirements, including those for file naming. The PLOS ONE style templates can be found at

Response for Point 1: Thank you for your insightful comments regarding PLOS ONE’s style requirements, including those for file naming. We appreciate your feedback and have carefully checked the style templates and file naming.

Point 2: Please note that PLOS ONE has specific guidelines on code sharing for submissions in which author-generated code underpins the findings in the manuscript. In these cases, we expect all author-generated code to be made available without restrictions upon publication of the work. Please review our guidelines at https://journals.plos.org/plosone/s/materials-and-software-sharing#loc-sharing-code and ensure that your code is shared in a way that follows best practice and facilitates reproducibility and reuse.

Response for Point 2: Thank you for clarifying the guidelines on code sharing. We confirm that all author-generated code underlying our findings will be made available without restrictions upon publication. To ensure adherence to PLOS ONE's best practices and facilitate reproducibility and reuse, we will provide clear documentation for the code, including any dependencies or software requirements. The code will be accessible via a stable, public repository, and we will include the relevant URLs or DOIs in our submission. We are committed to supporting transparency and reproducibility and will ensure that our code is shared in full compliance with PLOS ONE’s standards.

Point 3: We note that your Data Availability Statement is currently as follows: All relevant data are within the manuscript and its Supporting Information files.

Response for Point 3: Thank you for your detailed guidelines on the Data Availability Statement. We confirm that our submission contains all raw data necessary to replicate the results of our study, adhering to PLOS's definition of the minimal data set. Specifically:

The values underlying all reported means, standard deviations, and other relevant measures are included in the Supporting Information files.

The data used to construct all graphs and figures are available for review.

Points extracted from images for analysis, where applicable, are also included.

If additional information is required, we are open to uploading further Supporting Information files or depositing data in a recommended repository and providing the relevant access details. There are no ethical or legal restrictions on sharing de-identified data for our study, and all data are readily accessible in the submitted materials.

Point 4: We note you have included a table to which you do not refer in the text of your manuscript. Please ensure that you refer to Table 2 in your text; if accepted, production will need this reference to link the reader to the Table.

Response for Point 4: We apologise for any inconvenience caused by the omitted table that we do not refer to in our manuscript's text. We appreciate your feedback, and in response, we have revised the manuscript and referred to Table 2 in our text:

“As the groundwork for the credibility initiate process only to the Web of Science database and understanding that relevant studies on public sector performance could be omitted, we continue our approach through content analysis, gathering all the identified relevant scientific documents in Table 2.”

Point 5: Please review your reference list to ensure that it is complete and correct. If you have cited papers that have been retracted, please include the rationale for doing so in the manuscript text, or remove these references and replace them with relevant current references. Any changes to the reference list should be mentioned in the rebuttal letter that accompanies your revised manuscript. If you need to cite a retracted article, indicate the article’s retracted status in the References list and include a citation and full reference for the retraction notice.

Response to Point 5: We thank you for your comments and apologize for the inconvenience. Your recommendations helped us review our reference list to ensure that it is complete and correct. Due to adding a new section in the manuscript, namely the “Discussion” section, the list of references underwent a series of changes, including renumbering the order of references. We confirm that we have not retracted papers from the manuscript. Moreover, while revising our submission, we upload our figure files to the Preflight Analysis and Conversion Engine (PACE) digital diagnostic tool, https://pacev2.apexcovantage.com/. PACE to ensure that figures meet PLOS requirements.

COMMENTS FROM THE EXTERNAL PEER REVIEWERS

Manuscript ID: PONE-D-24-38083

Journal: PLOS ONE

Article title: Bridging Gaps in EU Public Sector Performance Assessment: An Analytical Approach Based on PLS Perspective

Reviewer: 1

Please find below specific answers to each comment/observation of Reviewer 1:

Point one: “This paper addresses an interesting topic related to the EU public sector performance assessment. I acknowledge the amount of work invested in preparing the manuscript and consider it significantly contributes to the scientific literature. I would only suggest to the author(s) to enhance the policy implications of own findings and limitations / future research directions, both could be augmented.”

Response for Point 1:

We thank you for your comments and apologize for the inconvenience. Your recommendations helped us enhance the policy implications of our findings, alongside the main limitations and future research directions of our research. Considering your valuable feedback, we reported within the manuscript accordingly (coloured in red) as follows:

“In addition, the implications of our results highlight the importance of a public sector performance index, both in terms of findings and limitations. Our main findings also sustain some critical paths of significance for measuring public sector performance. Nevertheless, through the existence of a public sector performance index, decision-makers have to be more responsible and communicative with citizens and stakeholders, ensuring that both parties use the resources efficiently. Moreover, the measurement of public sector performance identifies and highlights the areas in which government improvements are needed, thus guiding public policy adjustments and improving the delivery of quality public services to spur customer satisfaction. On the same lines, our main findings emphasise that having an index that measures the public sector performance facilitates the efficient allocation of resources, directing funding and efforts to programs that exert positive impact and multiple benefits within society. Moreover, transparency, public trust and involvement in government processes are stimulated. The public sector performance assessment can be considered a tool for operating decisions and improving public sector management practices, thus allowing comparisons between different sectors or regions and highlighting the exchange and implementation of best practices.

A distinct focus of our study is dedicated to identifying additional relevant indicators that can be included and measured in our analysis, as well as establishing new measurement procedures and criteria to pave the way for future research directions. This initiative is driven by the recognition that the current framework for evaluating public sector performance may benefit from a broader set of indicators that reflect the complexities of modern governance, particularly in relation to emerging issues such as environmental sustainability and climate change. We aim to continuously update our data by adapting our research framework to include new indicators and dimensions as they emerge. This dynamic approach allows our study to remain relevant and responsive to evolving societal and environmental challenges, particularly those related to climate change. Furthermore, we also consider developing strategies to combat climate change effects based on our analysis results. Therefore, we intend to expand the measurement of the public sector performance by proposing the creation of an index to measure the performance of the public sector and to have the role of an observatory at the European level, accompanied by permanent updates and unlimited availability. This multifaceted approach underscores our recognition of the critical intersection between public sector performance, digital transformation, and environmental sustainability.

The limitations of our research primarily stem from the challenges associated with data availability and the scope of the indicators chosen for analysis. These limitations can impact the overall accuracy and comprehensiveness of our findings, which are crucial for understanding the performance of the public sector in relation to digital transformation and climate change. Furthermore, in certain conditions, another limitation of our research may embody the relatively small group of pillars since there are multiple societal changes, thus not considering all the relevant components that may influence the measurement of the public sector performance. The lack of comprehensive data on specific indicators can lead to gaps in our understanding of how various dimensions of public sector performance are interrelated. Each pillar—be it digitalisation, governance, environmental impact, or societal factors—plays a crucial role in shaping overall performance. Insufficient data may obscure important relationships and nuances that could inform effective policy decisions. Another limitation arises from the relatively small group of pillars included in our analysis. While we aimed to capture key dimensions of public sector performance, societal changes are multifaceted, and many relevant components may not have been considered. This narrow scope can lead to an incomplete understanding of the factors influencing public sector performance.”

Thank you very much for the Reviewer’s 1 valuable suggestions and recommendations.

Reviewer: 2

Please find below specific answers to each comment/observation of Reviewer 2:

Point 1: “Title: quite long; try to reduce to one sentence (can be removed : An analytical approach based on PLS perspective)”.

Response for Point 1:

Thank you very much for your comment. Based on your recommendation, we have restructured the title to better reflect the focus of our study. The new title we propose is “Crossing Chasms: A PLS Approach to EU Public Sector Performance Assessment”.

Point 2: “Abstract: very large; try to introduce with the present results from the article”

Response for Point 2:

Thank you for your valuable feedback on the abstract. We appreciate your suggestion to introduce specific results. We believe that the revised abstract effectively summarises the study’s scope through the chosen methodology and its key findings while maintaining clarity. Accordingly, we concentrate on the present results from the article as follows:

“This paper examines the dynamics of public sector performance across European Union (EU) countries through a comprehensive methodological framework. This study introduces the European Public Sector Performance Index, a novel approach that employs Partial Least Squares (PLS) econometric modelling and cluster analysis to evaluate public sector performance from 2007 to 2021. By assessing performance across governance, social, and economic dimensions, the research captures the multifaceted nature of public sector efficiency in the EU. Our investigation reveals significant determinants of performance, including governance factors like Control of Corruption, Rule of Law, and Government Effectiveness, as well as economic indicators such as Inflation and social factors like Equity of access to healthcare services and Education Spending. These findings underscore the critical role of transparent governance, economic stability, and equitable social policies in enhancing public sector efficiency. Despite its reliance on secondary data and the PLS method, the study provides new methodological insights and empirical evidence on public sector performance, contributing to the literature with a holistic analysis that integrates digitalisation and well-being. This study’s holistic approach offers actionable insights for policymakers and stakeholders, emphasising the need for robust governance and equitable policies to improve public sector performance across the EU. The omission of certain societal components—such as economic conditions, demographic changes, or cultural factors—may result in a sk

---

## [Decision Letter · Decision Letter 1]

PONE-D-24-38083R1Crossing Chasms: A PLS Approach to EU Public Sector Performance AssessmentPLOS ONE

Dear Dr. Taran,

Thank you for submitting your manuscript to PLOS ONE. After careful consideration, we feel that it has merit but does not fully meet PLOS ONE’s publication criteria as it currently stands. Therefore, we invite you to submit a revised version of the manuscript that addresses the points raised during the review process.

**ACADEMIC EDITOR:**  Please consider the recommendations the reviewer made and change your paper accordingly.

We look forward to receiving your revised manuscript.

Kind regards,

Valentina Diana Rusu, PhD

Academic Editor

PLOS ONE

Journal Requirements:

Reviewers' comments:

Reviewer's Responses to Questions

**Comments to the Author**

1. If the authors have adequately addressed your comments raised in a previous round of review and you feel that this manuscript is now acceptable for publication, you may indicate that here to bypass the “Comments to the Author” section, enter your conflict of interest statement in the “Confidential to Editor” section, and submit your "Accept" recommendation.

Reviewer #2: All comments have been addressed

Reviewer #3: All comments have been addressed

Reviewer #4: (No Response)

2. Is the manuscript technically sound, and do the data support the conclusions?

Reviewer #2: Yes

Reviewer #3: Partly

Reviewer #4: Yes

3. Has the statistical analysis been performed appropriately and rigorously? 

Reviewer #2: Yes

Reviewer #3: Yes

Reviewer #4: Yes

4. Have the authors made all data underlying the findings in their manuscript fully available?

Reviewer #2: Yes

Reviewer #3: Yes

Reviewer #4: Yes

5. Is the manuscript presented in an intelligible fashion and written in standard English?

Reviewer #2: Yes

Reviewer #3: Yes

Reviewer #4: Yes

6. Review Comments to the Author

Reviewer #2: Journal: PLOS ONE

Article title: Crossing Chasms: A PLS Approach to EU Public Sector Performance Assessment

Manuscript ID: PONE-D-24-38083R1

Dear Author (s);

Dear Editor,

The manuscript has been revised for better interpretations according to the suggestions of the reviewer(s) by including the information required.

The author(s) change the interpretations, results, methodology, and conclusions accordingly, and therefore, the paper is much improved now. The author(s) reduces considerably the article, references, and diversifies the articles cited.

I recommend that this article be published in PLOS ONE.

Congratulations!

Reviewer #3: Congratulations on your acceptance! Your hard work has paid off, and this is a moment to celebrate. Well done!

Reviewer #4: The study "Bridging gaps in EU public sector performance assessment:An analytical approach based on PLS perspective", dealing with the complexity of European public sector performance by integrating the governance, economic and social pillars with an innovative methodology, provides a state-of-the-art perspective on the influence of transparent and accountable governance structures.

My specific recommendations and observations on this study are:

1. In the introductory section: a) Including recent data or statistics illustrating the current situation of public sector performance; these could provide a stronger justification for the necessity of the study and the chosen methodology b) Although the paper already includes several references, adding a critical discussion on the limitations of previous studies or different perspectives in the literature could more clearly emphasize the original contribution of this study

2. The "Materials and Methods" section is well structured and detailed, providing a good description of the empirical methods used in the study. However, a few improvements could be made to complete and clarify the methodological approach: a) Justification of the Selection of Indicators; b) Details on the validation of the PLS model.

4. The Results section presents a detailed and well-structured analysis of European public sector performance using composite indicators and PLS models. Aspects that could be improved or added to complete the section are a) Adding a short introduction reminding the purpose and importance of this analysis in the broader context of the study. This would help to anchor the results within the research objectives of the study; b) Why "Control of Corruption" is considered a major determinant compared to other variables and how these specific variables were chosen for the PLS model; c) It would be useful to include more direct comparisons with previous studies, highlighting how the current results contribute to the existing literature and what new insights they offer.

5. In the Conclusions Section the findings on digital economy factors are interesting, but it would be useful to explore further why these issues did not have a significant impact in the governance pillar and what this might mean for future research and policy implementation.

7. PLOS authors have the option to publish the peer review history of their article (what does this mean? ). If published, this will include your full peer review and any attached files.

**Do you want your identity to be public for this peer review?** For information about this choice, including consent withdrawal, please see our Privacy Policy .

Reviewer #2: No

Reviewer #3: No

Reviewer #4: No

---

## [Author Response · Author response to Decision Letter 2]

10 Mar 2025

Thank you for reviewing our submission and for the guidance provided. We have carefully addressed each point raised. We appreciate the journal's thorough review process and are happy to make any further adjustments required to meet PLOS ONE’s standards.

---

## [Decision Letter · Decision Letter 2]

Crossing Chasms: A PLS Approach to EU Public Sector Performance Assessment

PONE-D-24-38083R2

Dear Dr. Taran,

We’re pleased to inform you that your manuscript has been judged scientifically suitable for publication and will be formally accepted for publication once it meets all outstanding technical requirements.

Kind regards,

Valentina Diana Rusu, PhD

Academic Editor

PLOS ONE

Additional Editor Comments (optional):

Reviewers' comments:

Reviewer's Responses to Questions

**Comments to the Author**

1. If the authors have adequately addressed your comments raised in a previous round of review and you feel that this manuscript is now acceptable for publication, you may indicate that here to bypass the “Comments to the Author” section, enter your conflict of interest statement in the “Confidential to Editor” section, and submit your "Accept" recommendation.

Reviewer #1: All comments have been addressed

Reviewer #5: (No Response)

2. Is the manuscript technically sound, and do the data support the conclusions?

Reviewer #1: Yes

Reviewer #5: Yes

3. Has the statistical analysis been performed appropriately and rigorously? 

Reviewer #1: Yes

Reviewer #5: Yes

4. Have the authors made all data underlying the findings in their manuscript fully available?

Reviewer #1: Yes

Reviewer #5: Yes

5. Is the manuscript presented in an intelligible fashion and written in standard English?

Reviewer #1: Yes

Reviewer #5: Yes

6. Review Comments to the Author

Reviewer #1: After thoroughly reviewing the revised manuscript and the authors' responses, I am happy to report that they have effectively addressed the issues raised. The revisions have significantly improved the manuscript's clarity, coherence, and overall quality.

Specifically, the authors have successfully incorporated the suggested changes, clarified any ambiguous points, and strengthened the argument and evidence supporting their findings. The revised manuscript greatly enhances its contribution to the field, and I strongly recommend its publication in its current form.

Reviewer #5: Review for: Crossing Chasms: A PLS Approach to EU Public Sector Performance Assessment manuscript.

Dear Editor, Dear Authors,

Considering the previous reviews and suggestions, the revised version of the manuscript PONE-D-24-38083R2, entitled "Crossing Chasms: A PLS Approach to EU Public Sector Performance Assessment" is significantly improved, and it includes all the changes required.

Here are my personal remarks:

•Abstract section – clear and concise, it encompasses the aim of the paper, the main findings, together with the novelty of the paper.

•Introductory section – very well and clearly structured, especially for the research question as well as the novelty part, which is explained using five discussion points to prove the contribution in the field. This first section also provides a systematic bibliometric analysis as well as a brief literature review based on relevant papers for the topic of public sector performance. Moreover, all four research hypotheses are also presented at the end of this section.

•Material and methods section – good structure and explanation of the applied methods. The selection of the employed variables is well argued, as well as the data processing and the steps for the econometric framework.

•Results section – clear and structured presentation of the obtained results, both for the components of EU pillars as well as for the clustering part.

Only one remark here:

(lines 637-640)“Fig. 6 presents the results from a Partial Least Squares (PLS) model for the governance pillar of the European public sector, where the influence of 7 variables on governance is examined. The first five determinants collectively explain 90.14% of the variance within the model.”

The authors mention here (lines 637-640) that the influence of 7 variables is examined for the governance pillar, however, Fig. 6 provides the analysis of 9 determinants of governance. Please correct that in the manuscript.

•Discussion section – main results are emphasized again, together with a comparative and tabular presentation of the obtained performance scores, both on the governance, economic, social, and overall performance scores, which provides clarity. This section also compares the main results with other papers in the field.

•Conclusions – a bit too long; however they provide a very good explanation of the obtained results in relation to the research hypotheses. Implications, together with limitations are also stated.

I recommend that this manuscript be published, and I would like to congratulate the authors on their work throughout the manuscript's improvement stages.

Congratulations!

7. PLOS authors have the option to publish the peer review history of their article (what does this mean? ). If published, this will include your full peer review and any attached files.

**Do you want your identity to be public for this peer review?** For information about this choice, including consent withdrawal, please see our Privacy Policy .

Reviewer #1: No

Reviewer #5: No

---

## [Editor Report · Acceptance letter]

PONE-D-24-38083R2

PLOS ONE

Dear Dr. Taran,

I'm pleased to inform you that your manuscript has been deemed suitable for publication in PLOS ONE. Congratulations! Your manuscript is now being handed over to our production team.

Kind regards,

on behalf of

Dr. Valentina Diana Rusu

Academic Editor

PLOS ONE